# Advances in the Chemistry, Analysis and Adulteration of Anthocyanin Rich-Berries and Fruits: 2000–2022

**DOI:** 10.3390/molecules28020560

**Published:** 2023-01-05

**Authors:** Bharathi Avula, Kumar Katragunta, Ahmed G. Osman, Zulfiqar Ali, Sebastian John Adams, Amar G. Chittiboyina, Ikhlas A. Khan

**Affiliations:** 1National Center for Natural Products Research, University, MS 38677, USA; 2Division of Pharmacognosy, Department of BioMolecular Sciences, School of Pharmacy, University of Mississippi, University, MS 38677, USA

**Keywords:** anthocyanins, chemistry, stability, analytical techniques, fruit and berries, health benefits

## Abstract

Anthocyanins are reported to exhibit a wide variety of remedial qualities against many human disorders, including antioxidative stress, anti-inflammatory activity, amelioration of cardiovascular diseases, improvement of cognitive decline, and are touted to protect against neurodegenerative disorders. Anthocyanins are water soluble naturally occurring polyphenols containing sugar moiety and are found abundantly in colored fruits/berries. Various chromatographic (HPLC/HPTLC) and spectroscopic (IR, NMR) techniques as standalone or in hyphenated forms such as LC-MS/LC-NMR are routinely used to gauge the chemical composition and ensure the overall quality of anthocyanins in berries, fruits, and finished products. The major emphasis of the current review is to compile and disseminate various analytical methodologies on characterization, quantification, and chemical profiling of the whole array of anthocyanins in berries, and fruits within the last two decades. In addition, the factors affecting the stability of anthocyanins, including pH, light exposure, solvents, metal ions, and the presence of other substances, such as enzymes and proteins, were addressed. Several sources of anthocyanins, including berries and fruit with their botanical identity and respective yields of anthocyanins, were covered. In addition to chemical characterization, economically motivated adulteration of anthocyanin-rich fruits and berries due to increasing consumer demand will also be the subject of discussion. Finally, the health benefits and the medicinal utilities of anthocyanins were briefly discussed. A literature search was performed using electronic databases from PubMed, Science Direct, SciFinder, and Google Scholar, and the search was conducted covering the period from January 2000 to November 2022.

## 1. Introduction

Anthocyanins are polyphenolic plant constituents and are the most commonly occurring flavonoids in many fruits, especially berries, and in several vegetables. Anthocyanins consumption is approximately nine times higher than other nutritional flavonoids in certain food products. Most anthocyanins are water-soluble glycosides and chemically are derivatives of the 2-phenylbenzopyrylium or flavylium salts. Anthocyanins have a distinctive ability to form flavylium cations and acquire different colors, from red to blue or violet, depending on the pH of the medium [1]. The term anthocyanin refers to glycoside, while anthocyanidin refers to aglycone. Approximately 23 aglycones have been identified and characterized; however, only six are widely distributed among plants, namely cyanidin, delphinidin, malvidin, pelargonidin, peonidin, and petunidin [1,2]. Anthocyanins are more stable in acidic solutions (pH 1–3), where they exist as flavylium cations. These colored constituents occur in various berries and fruits of the genera representing *Prunus*, *Vaccinium*, *Vitis*, *Ribes*, *Morus*, *Fragaria*, *Aronia*, and *Rubus* [3]. The composition of the anthocyanin content (the chemical profile of anthocyanin) depends on the cultivar, maturity, collection season, geographic region, and other factors [4,5].

Taxonomically, a fruit is defined as a berry if the outside wall of the ovary from a single flower matures into an edible, fleshy pericarp. Drupes, fruit having a hard layer (endocarp) surrounding the seed, are also often called berries. Many berries, drupes, pomes (e.g., apples), and aggregate fruits (e.g., strawberries, raspberries, etc.) are rich in anthocyanins (Figure 1). This review focuses on all types of anthocyanin-rich berries and fruits listed in Table 1.

The therapeutic and health-promoting effects of anthocyanins are attributed to their antioxidant and anti-inflammatory activities. Moreover, anthocyanins contribute to the alleviation of the severity of diabetes, obesity, and cancers via inhibition of the NF-κB mediated inflammatory pathways. Nevertheless, substantial disparities in the effects of anthocyanins have been observed, primarily due to the structural diversity of anthocyanins [6]. Numerous studies, including in vitro, animal models, and human clinical trials, indicated that anthocyanidins and anthocyanins possess a wide range of health-promoting and pharmacological effects, including antioxidant, attenuation of metabolic syndromes and cardiovascular diseases, enhancing immune system, improvement of cognitive function, antimicrobial activities, and improving visual and neuroprotective effects. Anthocyanins showed the ability to attenuate reactive oxygen species (ROS) and reactive nitrogen species (RNS) in human, animal, and in vitro studies [7,8,9]. Findings from in vitro, animal studies, and human trials substantiate the antioxidant effects of anthocyanins [10,11]. In addition, anthocyanins show cytoprotective, antitumor, anti-obesity, and lipid-lowering effects. Epidemiological evidence indicated a direct correlation between anthocyanin intake and a lower incidence of chronic and degenerative disorders [12].

Evidence at multiple levels substantiates the role of anthocyanins and anthocyanin-rich fruits and vegetables in regulating several mechanistic pathways that culminate in enhancing human health and preventing or delaying the onset of chronic medical conditions. Seemingly, anthocyanins are safe, given their recently reported safety profile [13,14]. In this literature review, the major aspects of the polyphenolic natural products, anthocyanins, are discussed in view of the data presented in the literature. These aspects include chemistry, and analytical approaches, including liquid chromatography coupled with mass spectrometry (LC-MS), nuclear magnetic resonance (NMR), and high-performance thin-layer chromatography (HPTLC), which represent the most popular techniques utilized for qualitative and quantitative analysis of anthocyanins, in addition to infrared spectroscopy (IR) and capillary electrophoresis (CE). Botanical sources of anthocyanin-rich berries and fruits, as well as adulteration of genuine berries and fruits and detection of economically motivated adulterants, will also be covered.

## 2. Occurrence, Chemistry, and Stability of Anthocyanins

Most berries possess simple anthocyanin profiles with at least 80% of the total anthocyanins represented by a limited number of compounds except for blueberries, and bilberries where more complex profiles were observed. Anthocyanins in berries are mainly glycosides of cyanidin, delphinidin, peonidin, pelargonidin, malvidin, and petunidin. They may be glycosylated with glucose, galactose, arabinose, or rutinose, and to a lesser extent with xylose in chokeberry, sambubiose in elderberry, and sophorose in raspberry.

Cyanidin is the most common anthocyanidin. It is generally found as cyanidin-3-glucoside, represented as Cy-3-glc. Cyanidin derivatives are the major anthocyanins in black elderberry (glucoside and sambubioside), black chokeberry (galactoside, arabinoside, xyloside, and glucoside), blackberry (glucoside), red raspberry (sophoroside), açai berry (rutinoside, glucoside), sweet cherry (rutinoside, glucoside), plum (rutinoside), and red apple peel (galactose). The main anthocyanin in black raspberry is cyanidin-3-(6-*p*-coumaroyl)-glucoside, represented as Cy-3-(6-*p*-cou)-glc [15]. Table 2 shows the occurrence of anthocyanins in various berry and fruit varieties based on analytical methodologies.

The anthocyanins, a major flavonoid group, are water-soluble vacuolar pigments consisting of an aglycone (anthocyanidin), sugar unit/s, and in some cases, acyl group(s). Over 600 naturally occurring anthocyanins have been reported so far varying in (1) the number and position of hydroxyl and methoxyl groups on the basic anthocyanidin skeleton; (2) the identity, number, and positions at which sugars are attached; and (3) the extent of sugar acylation and identity of the acylating agent [16]. Although about 23 anthocyanidins have been found in nature, only six of them account for ~90% of the total distribution of those six beings: cyanidin (Cy) (50%), delphinidin (Dp) (12%), peonidin (Pn) (12%), pelargonidin (Pg) (12%), petunidin (Pt) (7%), and malvidin (Mv) (7%). These most common anthocyanidins occur as glycosides having different substitutions on their B-rings [17]. The following four classes of anthocyanidin glycosides are common: 3-monosides, 3-biosides, 3,5-diglycosides, and 3,7-diglycosides. Many anthocyanins containing mono glucose unit (3-glycosides) occur more frequently than 3,5-diglycosides. The most widespread anthocyanin is cyanidin 3-glucoside [17]. Glucose (glc), galactose (gal), arabinose (ara), rhamnose (rha), and xylose (xyl) are the most common sugars that are bonded to anthocyanidins as mono-, di-, or tri-saccharide forms. The most common acyl groups are coumaric, sinapinic, malonic, acetic, caffeic, ferulic, succinic, oxalic, malic, and *p*-hydroxy benzoic acids. The lack of commercially available anthocyanin standards has limited quantitative analyses of many anthocyanins in various matrices. The sugar moieties mainly attach to 3-position on the C-ring or the 5, 7-position on the A-ring. Differences in the environmental and growing conditions of the berries and fruits, as well as the genetic factors and the harvesting season, lead to variation in the composition of anthocyanins from one cultivar to another, and even within the same species but from a different geographic region. For purified anthocyanins, additional factors including drying methods of the plant raw material, extraction, and processing contribute to variability of anthocyanin content and composition [18]. The color and color intensity of anthocyanins are affected by the number of hydroxyl and methoxyl groups [9] attached to these aromatic rings. Often, the number of hydroxyl or methoxyl groups dictates the final color—the bluish shade associated with a higher number of hydroxyl groups and the reddish shade associated with more methoxyl groups. The sugar moieties are usually connected to the anthocyanidins through *O*-linkages, where both 3-*O*-glycosides and 3,5-*O*-diglycosides are dominant, but also *C*-glycosylation at the positions C7, C3′, and C5′ of the anthocyanidin molecule has been reported [19].

### Stability Considerations

Anthocyanins are reported to be prone to oxidation, with light, temperature, water content, oxygen, enzymes, co-pigments, ascorbic acid, proteins, metal ions and pH affecting their stability [20,21,22,23]. The instability of anthocyanins’ structure under the influence of temperature, direct sunlight, pH and solvents is a main problem in analytical investigations of anthocyanins.

Anthocyanins may exist in at least four different pH-dependent structural isoforms, namely, red flavylium cation, colorless hemiketals, blue quinoid bases, and pale-yellow retro-chalcones, arising at pH 1–3, 4–5, 6–8, and 7–8, respectively [24] (Figure 2). At more alkaline pH values, anthocyanins have consistently been shown to degrade to their constituent phenolic acids, where delphinidin, cyanidin, and pelargonidin degrade to form gallic acid, protocatechuic acid, and 4-hydroxybenzoic acid, respectively [25,26]. With respect to molecular structure, some anthocyanins are more stable than others. Generally, increased hydroxylation decreases stability, whereas increased methylation increases it.

Due to possible hydration, the anthocyanins generally begin to lose color at pH > 2.5–3.0 in an aqueous solution. To stabilize the color of anthocyanins, two mechanisms involved are [20] complexation by metal ions using aluminum, magnesium or ferric ions and forming self-assembling supramolecular complexes that incorporate colorless “co-pigment” molecules such as flavones. This complex-mediated co-pigmentation results in blue or purple colors [27] and [21] complexation of the flavylium cation in the absence of metal ions by colorless co-pigment molecules such as hydroxylated benzoic and cinnamic acids, hydroxy-flavones and other polyphenols [27]. The basic role of co-pigments is to protect the colored flavylium cation from nucleophilic attack by water.

## 3. Preparation of Anthocyanin Samples for Analysis

Anthocyanins are unique among plant phenolics because they can be present in plant tissues in different structural isoforms. They predominantly exist in flavylium cation form at low pH, giving a reddish color in aqueous solutions. At higher pH, the flavylium cation is converted into other species, some uncolored with that conversion being virtually irreversible under certain conditions. The flavylium cation form is red and stable in a highly acid medium. This chemical feature of anthocyanins has probably led to a worldwide use of solvents containing mineral or organic acids for the extraction of anthocyanins. In neutral solutions, anthocyanins exist as noncharged quinoidal forms [28,29].

Anthocyanins are commonly extracted with methanol; however, solvents such as ethanol, acetonitrile, acetone, and water containing small amounts of hydrochloric, formic, acetic or phosphoric acid may be used. Adding acid lowers the pH of the solution and prevents the degradation of the non-acylated anthocyanin pigments. However, increasing the acid may cause partial or total cleavage of the acyl moieties of the acylated anthocyanins in some plants. There is no universal sample pretreatment method available for all kinds of samples and sampling, sample preservation and preparation should be well considered and documented for the rigor and reproducibility of analytical findings. However, the extraction and analysis protocols are difficult to accomplish because of the structural diversity of anthocyanins and their susceptibility to heat, pH, metal complexes, and co-pigmentation.

To determine the target analytes and overall quality, the composition and content of glycosides or aglycones are essential. When dealing with plants, food products, and biological materials, the conjugates are usually searched for, whereas in other instances, it is necessary to carry out preliminary hydrolysis, e.g., an enzymatic or chemical (acidic or alkaline) treatment. Intentional hydrolysis for obtaining the aglycones of some flavonoids or deriving fatty acids to their corresponding esters is sometimes intentionally incorporated into the extraction process. A variety of modern techniques have been developed for this purpose, including ultrasound-assisted extraction (UAE), solid phase extraction (SPE), accelerated solvent extraction (ASE), countercurrent chromatography (CCC), microwave-assisted extraction (MWE), supercritical fluid extraction (SFC), pressurized hot water extraction, and high hydrostatic pressure extraction. The choice of an extraction method should be based on maximizing the anthocyanin recovery with the minimum amount of non-anthocyanin components and minimal degradation or alteration of the native anthocyanins. Haffels et al. [30] investigated the influence of ASE and UAE using two different solvent compositions on the anthocyanin profile of 27 samples representing bilberries, lowbush blueberries, and American cranberries. Besides differences in total anthocyanin content in the extracts, significant deviations (*p* ≤ 0.05) in the individual anthocyanin concentration were observed, resulting in altered yields and peak ratios. It was revealed that the chemical profile variations induced by the extraction methods are similar to those attributed to the geographic and climatic differences. This study showed that the sample preparation procedures for analyzing the chemical composition of anthocyanins at the qualitative and quantitative levels are crucial for the comparability of *Vaccinium* berries’ authenticity. Considering the results obtained in the current investigation, the extraction method should also be reviewed for authenticity data.

## 4. Analytical Techniques including Identification and Quantification of Anthocyanins

The search for new ingredients of natural origin, including anthocyanins, and the study of their bioactivities have been ongoing for several decades due to the growing demand for natural products with beneficial health properties. Separation of anthocyanins is challenging in many aspects. The fruits mentioned above contain a group of anthocyanins with very similar structures. They belong to the large group of phenolic compounds, and, however, most of these bioactive compounds degrade under exposure to high temperatures, oxygen, and light, conditions that could decrease the nutritional value during the processing, storage, and product distribution. In this regard, identifying methods capable of exerting efficient quality assurance and quality control of the final products is essential for delivering the bioactive components in an unaltered form at the intended concentrations.

There are several analytical techniques used for anthocyanin identification and characterization in different matrices such as: Nuclear Magnetic Resonance (NMR), Near Infrared or Far Infrared spectroscopy (NIR/FIR), Thin Layer Chromatography/High-Performance Thin Layer Chromatography (TLC/HPTLC), Capillary Zone Electrophoresis (CZE), Micellar Electro-Kinetic Chromatography (MEKC), and High-Performance Liquid Chromatography or Ultra-High Performance Liquid Chromatography with Ultra-Violet or Mass Spectrometry (LC-UV/Vis/MS). The cost of these methods is very high because of the complicated equipment. Paper and/or thin-layer chromatography and UV-Vis spectroscopy have traditionally been used for the identification of anthocyanins. Capillary zone electrophoresis, a hybrid of chromatography and electrophoresis, is gaining popularity for the analysis of anthocyanins; however, liquid chromatography (LC) has become the standard method for identification and separation in most laboratories and is used for both preparative and quantitative analysis. LC-MS and NMR spectroscopy are possibly the most powerful methods for the characterization and structural elucidation of anthocyanins available. Currently, the most satisfactory method for mixture analysis is the multistep separation, isolation, and quantification by LC- UV/Vis with peak identification using MS and high-field NMR [31]. In Table 3 and Table 4, several examples of studies employing these techniques have been collected.

The database or search engines used were SciFinder, PubMed, Science Direct, and Google Scholar. Search terms were related to ‘anthocyanins’ and ‘fruit/berry name’ and ‘analytical method’, restrictions were applied concerning the English language, only, and publication dates (1 January 2000–30 November 2022). Conference abstracts, letters to the editor, conference proceedings, literature reviews, and systematic reviews were excluded. Selection criteria included were: (1) only fruit or berry parts are used in the study, no other plant parts (leaf, root, stem) were used, and (2) in most cases the processed food or dietary supplements were excluded in the analytical method sections but discussed under section ‘adulteration issues’.

### 4.1. Nuclear Magnetic Resonance (NMR)

Compared to chromatographic analyses, NMR is less sensitive, but it is non-destructive, simpler, shorter in sample preparation, more reproducible, and capable of simultaneous detection of many organic components and profiling a broad range of secondary metabolites in complex mixtures. NMR techniques have been used to identify anthocyanins and study their exact structural characteristics, and to establish their mechanisms of action, which can lead to better applications of these compounds as functional ingredients [32].

The NMR approach was used for berries/fruits such as chokeberry, bilberry, blackcurrant, apple, black raspberry, grape, blueberry, pomegranate, and mulberry. The studies report the use of NMR spectroscopy, either ^1^H or ^13^C using correlation spectroscopies COSY or TOCSY and HSQC in CD_3_OD. NMR techniques coupled with chemometrics, viz., Principal Component Analysis (PCA), have recently been applied to metabolic profiling and determining the geographical origin of plant materials and food sources.

Wyzgoski et al. [33] investigated black raspberry extracts with high-field ^1^H NMR combined with multivariate statistical analysis to build a model using total monomeric anthocyanin (TMA), antioxidant capacity data from ferric-reducing antioxidant power (FRAP), and 2,2-diphenyl-1-picrylhydrazyl (DPPH) assays. Anthocyanins, Cy 3-rut, Cy 3-xyl-rut, and Cy 3-glc, were significant contributors to the variability in assay results. De la Cruz et al. [34] investigated four *Vitis* genotypes using LC–ESI–MS^2^ and LC–NMR–Foxy experiments. In this study, the authors identified and quantified 30 anthocyanins including two new *cis*-*p*-coumaroyl derivatives in two *Vitis* species, by LC-MS and LC-NMR experiments. *V. candicans* and *V. doaniana* showed substantial differences in their anthocyanin profiles. The identification of *cis* and *trans* isomers of *p*-coumaroyl-glucoside and *p*-coumaroyl-diglucoside was based on coupling constant values in the LC–^1^H-NMR spectra. The higher coupling constant (16 Hz) and smaller one (10 Hz) were characteristic for trans and cis isomers, respectively. Park et al. [35] used Hierarchical Cluster Analysis (HCA) dendrograms derived from ^1^H-NMR data grouped fruit samples according to species and geographic origin, and these matched Random Amplified Polymorphic DNA (RAPD) data of leaf samples, whereas the HCA dendrogram using flavonoid, phenolic acid, flavonol, and anthocyanin contents did not match the RAPD data. Thus, these results suggest that NMR-based metabolic profiling is a useful method to differentiate black raspberry fruit species and geographical origins. Capitani et al. [36] used aqueous and organic extracts of blueberries for untargeted NMR metabolite profiling as well as targeted NMR characterization of anthocyanins using a solid phase extraction. Five anthocyanins were identified as Mv-3-glc, Mv-3-gal, Dp-3-glc, Dp-3-gal, and Pt-3-glc. Goulas et al. [37] developed an NMR- based methodology for quantitative and qualitative characterization of metabolites including anthocyanins in a single run without the involvement of separation/purification. Diagnostic peaks for anthocyanins were located at 8.2–8.6 ppm for the H-4 proton. Hosoya et al. [38] qualified and quantified anthocyanins by multivariate statistics on ^1^H-NMR spectra in 13 kinds of berries. Each anthocyanin of cyanidin and pelargonidin glycosides was identified in the ^1^H-NMR spectra using the specific signals around 8.80–9.20 ppm. Using the ^1^H-NMR spectra, Principal Component Analysis (PCA) revealed that 13 kinds of berries could be distinguished. This NMR-based molecular species method may be applied to controlling the quality of products and the screening of unique types of berries. Tian et al. [39] investigated the phenolic compounds of berries and the leaves of various berry plant species. These collections of berries and leaves were separately extracted with aqueous ethanol and analyzed using UPLC-DAD-ESI-MS, HPLC-DAD, and NMR. Full scan NMR was performed on all extract samples in order to provide an overall profile of the metabolites. Overall, the aromatic region (all compounds with a benzene ring) was richer in berries than in leaves and signals at 6.9–7.1 ppm was typical for proton NMR at C6 and C8 of flavonoids and proanthocyanidins.

Zielińska et al. [40] evaluated the anthocyanin composition of chokeberry showing the presence of the structure of cyanidin galactoside and cyanidin arabinoside during the second stage of fruit ripening using NMR spectroscopy (1-D and 2-D) and HPLC-DAD. The ^1^H-NMR spectrum showed distinct signals in the aromatic region at 6.5–9.0 ppm which did not appear in the spectra recorded for the green fruit extracts (first growth stage). Turbitt et al. [41] implemented ^1^H-NMR-based chemometrics for cranberry supplements and whole cranberry powder to characterize variations in total proanthocyanidins and anthocyanins. Hasanpour et al. [42] investigated and compared the metabolite profiles of pomegranate ecotypes grown in eight geographical origins of Iran using 1D ^1^H-NMR spectroscopy associated with additional 2D NMR techniques. Delphinidin, cyanidin, and pelargonidin 3,5-diglucosides were identified. Multivariate statistical analyses, PCA, and Orthogonal Partial Least Squares-Discriminant Analysis (OPLS-DA) combined with NMR were applied to reveal differences and ecotypic diversity among eight geographical origins of Iran. Li et al. [43] explored the effect of drying conditions on mulberries at four drying temperatures: 40, 50, 60, and 70 °C. Low Field NMR (LF-NMR) and Magnetic Resonance Imaging (MRI) detected the water state and distribution during the drying process. The total anthocyanin content was highest when the mulberries were dried at 50 °C and lowest at 70 °C. Notably, higher correlations between LF-NMR parameters and quality properties were found by Partial Least Squares Regression (PLSR), with the analysis results being credible.

### 4.2. Infrared Spectroscopy (IR)

IR is a nondestructive method and could be beneficial in reducing the analysis time and labor costs and avoiding solvents while providing reliable and robust information on the anthocyanin composition. Near Infrared (NIR) spectroscopy has increased in acceptance having a primary potential advantage of using intact samples without sample preparation. Though the predominant methods of analysis of anthocyanins and anthocyanin-rich berries and fruits are based on LC-MS, HPLC, and ^1^HNMR techniques, the literature showed that a few methods of analysis employed Fourier Transform IR (FTIR), with NIR spectroscopy being the most popularly used technique. Other applications included IR-assisted freeze drying to keep the quality of anthocyanin-rich berries and maintain the nutritional quality of these fruits [44], and use of both the visible (Vis) and near infrared (NIR) (400–2500 nm) spectra to investigate the effect of homogenization and storage on different characteristics of red grapes, particularly the color and anthocyanin content. Other applications included utilization of Far-infrared Radiation heating assisted Pulsed Vacuum Drying (FIR-PVD) as an efficient drying method for blueberries. The objective of this method was to efficiently dry blueberries while maintaining the original color and stability of their anthocyanin content [45]. In another study, 168 samples of raspberries were obtained from plants at different ripening stages covering the whole period from the un-ripened to the fully ripened stage. The analysis revealed significant variation in the three determined parameters among the different samples at different maturation stages indicating the anthocyanin content is dependent on the maturity stage of the harvested berries. This method can be rapid and accurate, and can be used for quality control [46].

Red grape “musts” are freshly crushed grape juice and contain the skin, seeds and fruit stalks. In the current method of analysis, the anthocyanin content of the musts of red grapes, 12 anthocyanins (five non-acylated, three acetylated, three *p*-coumaroylated and one caffeoylated 3-*O*-glucosides) were quantitatively analyzed by Fourier Transform Mid-IR (FT-MIR) combined with partial least squares regression, using 257 samples of red grape musts obtained from three harvests. The method was proven to be quick and simple and can be used as a parameter of the quality control of harvested grapes [47]. Diffuse reflectance infrared Fourier transform spectroscopy (1700–1500 cm^−1^), was used for quantitative analysis of anthocyanin content in three sweet cherry varieties (Vogue, Van, and Hardy Giant) along with chemometrics for statistical validation. Hernández-Hierro et al. [48] evaluated near-infrared (900–1700 nm) hyperspectral imaging to determine anthocyanins in intact grapes of Syrah and Tempranillo during ripening. Inácio et al. [49] evaluated the potential of near-infrared reflectance spectroscopy and multivariate calibration to determinate total anthocyanin content in intact açai fruit without sample preparation. Several multivariate calibration techniques, including PLS, iPLS, SPA, GA, and outlier detection were conducted and compared to determine the best performing models indicating that the model developed by NIR spectroscopy for TAC can be used as an alternative to UV-Vis measurements. Beghi et al. [50] studied apple fruit quality using Vis/NIR. Chen et al. [51] demonstrated the capability of near-infrared (900–1700 nm) hyperspectral imaging in predicting the changes of anthocyanin content in wine grape skins during ripening. Martínez-Sandoval et al. [52] evaluated the heterogeneity of anthocyanins during red grape ripening using a near infrared hyperspectral imaging device (900–1700 nm). Yahui et al. [53] studied NIR based on chemometrics for the quality control of black goji berries sourced from different origins and was able to provide an accurate prediction of anthocyanin in black goji berries. He also found that one cluster separated from others and was identified as being adulterated. Zhang et al. [54] focused on the predictive models for total iron reactive phenolics, anthocyanins, and tannins in grape skin and seed for five red wine grape cultivars (Cabernet Sauvignon, Shiraz, Pinot Noir, Marselan, Meili) during ripening based on near-infrared (977–1625 nm) hyperspectral imaging technology. For grape skins, the Marselan cultivar shows the highest reflectance, whereas the Meili cultivar shows the lowest. Zhang et al. [55] developed near-infrared (874–1734 nm) hyperspectral imaging to determine total phenolics, total flavonoids and total anthocyanins in dry black goji berries. Machine learning via Convolutional Neural Networks (CNN) was applied in prediction of chemical compositions. Gales et al. [56] developed an NIR quantification model for the anthocyanins present in whole fresh raspberries.

### 4.3. High Performance Thin Layer Chromatography (HPTLC) Using UV and MS

High-performance thin layer chromatography (HPTLC) has gained wide popularity as a versatile instrumental technique operating with a standardized methodology for identification, quantification, and fingerprinting of botanical products. It is characterized by possessing several advantages including automation, scanning, optimization, reliability, accuracy, and reproducibility. HPTLC is a cost-efficient separation technique with selective detection of compounds belonging to different phytochemical classes and minimum sample preparation. Additionally, it can be hyphenated with other techniques such as mass spectrometry or HPLC to enhance its efficiency. HPTLC has been widely utilized for the analysis of polyphenols such as anthocyanins and flavonoids, simple phenols, and phenolic acids. The introduction of new stationary phases improved the resolution of complex mixtures of natural products extending the analysis capacity to more classes of phytochemicals.

Lambri et al. [57] developed a qualitative and quantitative method for red wine pigments. The substances separated by HPTLC were identified by HPLC-DAD at 520 nm. Rumalla et al. [58] developed a validated HPTLC method for quantifying two major anthocyanins, viz., Cy-3-*O*-rut and Cy-3-*O*-glc, from the berries of açai and dietary supplements. Kruger et al. [59] developed a quantitative method for main anthocyanin measurement in pomace, feed, juice, and wine using HPTLC with unknowns characterized by coupling with ESI-MS. The pattern was characteristically different between plant sources, e.g., elderberry juice differed from blackcurrant juice. Cretu et al. [60] used high-performance thin-layer chromatography (HPTLC) with bioassay to discern the specific anthocyanin content of five berry extracts, including açai berry, bilberry, blueberry, chokeberry, and cranberry. Bilberries showed the highest anthocyanin content (59.5%), followed by chokeberry (15.5%), blueberry (2.5%), açai berry (0.14%), and cranberry (0.08%). They also collected MS spectra to confirm the findings. In the same study, bacterial assay with *A. fischeri* was used to evaluate the samples for bioluminescence. It was determined that malvidin presented a strong response within the various samples and this response became stronger over the 30 min testing window. David et al. [61] evaluated the authenticity of 12 commercially available red fruits (raspberry, strawberry, blueberry, rose hip) using HPTLC fingerprinting. The HPTLC fingerprints obtained after the derivatization with Natural Product/Polyethylene Glycol (NP/PEG) reagents indicated different characteristic colored zones allowing a clear differentiation between all samples. In another similar study, Craciun et al. [62] conducted the quantification of four anthocyanins (Mv 3-glc, Cy 3-glc, Pn 3-glc and Dp 3-glc) from cranberry, blueberry, bilberry, chokeberry and açai berry extracts and by coupling with mass spectrometry (TLC-MS) confirmed the presence of cyanidin in chokeberry and açai berry extracts. Kruger et al. [63] also investigated anthocyanin patterns of fresh and dried elderberry using HPTLC-ESI-MS (Table 3). The anthocyanin content was higher in berries of cultivars than in wild-growing plants. They found that the specific compounds (Pn-3-glc, Mv-3-glc, Dp-3-glc and Cy-3-glc) demonstrated radical scavenging and that the anthocyanidins in the cranberry powdered extract were active. Koss-Mikołajczyk et al. [64] showed the relationship between the content of bioactive compounds and mutagenic activity of elderberry fruit at different stages of ripeness. Significant differences in the profiles of TLC, HPLC and antioxidant activity (ABTS, DPPH, and FC tests) were observed for studied elderberry samples. The more ripened the fruit at the time of harvest, the higher was the content of anthocyanins (an increase from 0 to 7.8 mg/g dry weight) and antioxidant activity of the extracts (about 5-fold increase). Cyanogenic glycosides were not detected at any stage of ripeness. Bernardi et al. [65] demonstrated the synergy between HPTLC-MS and HPLC-DAD methods for investigating pomace and seed samples from white and red *Vitis vinifera* cultivars. This study showed that Mv-3-glc was the most abundant. Finally, many of these researchers used DPPH reagent to analyze the samples for radical scavenging activity [59,60,61,62]. HPTLC also allows researchers to analyze anthocyanins and anthocyanidins using a single sample and provides for the cost-effective use of mass spectrometry instrumentation. However, HPLC remains the preferred method for evaluating for minor anthocyanins (Table 3).

**Table 3 molecules-28-00560-t003:** HPTLC and CE analysis of fruit/berries for analysis of anthocyanins.

No.	Identification Technique (Year)	Source	Extraction Solvent	Compounds	Activity	Ref
1	HPTLC (2003)	Skin of *V. vinifera*	MeOH-HCl (99.1:0.1) for 16 h agitation at 25 °C	Glycosides, acetylglycosides and coumaroylglycosides of 5 (Dp, Mv, Cy, Pn, Pg) main anthocyanins	-	[57]
2	HPTLC fingerprinting (2003)	Red fruits and teas (raspberry, straw-berry, blueberry, rose hip, wild berry)	acidified ethanol (HCl 0.1% *v*/*v*)	-	-	[61]
3	HPTLC (2012)	Açai berry and supplements	Methanol-10% aq. formic acid (9:1)	Cy-3-glc, Cy-3-rut	-	[58]
4	HPTLC-ESI-MS (2013)	Grape skin, pomace, Grape juice, wine		Mv 3-glc, Cy 3-glc, Pn 3-glc, Dp 3-glc, Pg-3-glc and diglc of Mv	-	[59]
5	HPTLC–Vis–MS (2014)	bilberry, blueberry, chokeberry, açai berry and cranberry	mixture of methanol and hydrochloric acid, 25%, 4:1, *v*/*v*	Mv 3-glc, Cy 3-glc, Pn 3-glc, Dp 3-glc	-	[60]
6	HPTLC-UV/Vis and HPTLC-ESI-MS (2015)	Fresh and dried elderberry	Acidified methanol	Cy-3-sam and Cy-3-glc	-	[63]
7	HPTLC (2015)	cranberry, blueberry, bilberry, chokeberry and açai berry	Acidified methanol (4:1)	Mv 3-glc, Cy 3-glc, Pn 3-glc, Dp 3-glc	Antioxidant activity	[62]
8	CE-UV (2003)	Grapes skin and wine	5% *v*/*v* of formic acid in methanol	My-3, 5-diglc, My-3-glc, My-3-gal, Pg-3-glc, Cy-3, 5-diglc, Cy-3-gal	-	[66]
9	CE-UV (2000)	Wild-type blueberry (bilberry)	3% aq.TFA	Mv-3-glc, Pn-3-glc, Pt-3-glc, Cy-3-glc, Mv-3-gal, Pt-3-gal, Dp-3-glc, Cy-3-gal, Dp-3-gal, Cy-3-ara, unknown	-	[67]
10	CE-UV (2001)	Wild-type blueberry (bilberry)	1% TFA	Mv-3-glc, Pn-3-glc, Pt-3-glc, Cy-3-glc, Mv-3-gal, Pt-3-gal, Dp-3-glc, Cy-3-gal, Dp-3-gal, Cy-3-ara, unknown 1-2	-	[68]
11	CE-UV (2004)	Wild-type blueberry (bilberry)	3% aq.TFA	Mv-3-glc, Pn-3-glc, Pt-3-glc, Cy-3-glc, Mv-3-gal, Pt-3-gal, Dp-3-glc, Cy-3-gal, Dp-3-gal, Cy-3-ara, Mv-3-ara, Pn-3-ara, Pn-3-gal, Pt-3-ara	-	[69]
12	CE-UV (2004)	Wild-type blueberry (bilberry)	[1] 100 mM of AAPH at pH 5.6 (0.1 M phosphate buffer) [2] H_2_O_2_ and t-BuOOH	Mv-3-glc, Pn-3-glc, Mv-3-gal, Pt-3-glc, Pn-3-gal, Cy-3-glc, Dp-3-glc, Pt-3-gal, Cy-3-gal, Dp-3-gal, Cy-3-ara, Dp-3-ara	-	[70]
13	CE-UV (2004)	Wild-type blueberry (bilberry)	1% aq.TFA	Mv-3-glc, Pn-3-glc, Pt-3-glc, Cy-3-glc, Mv-3-gal, Pt-3-gal, Dp-3-glc, Cy-3-gal, Dp-3-gal, Cy-3-ara, Mv-3-ara, Pn-3-ara, Pn-3-gal, Pt-3-ara, Dp-3-ara	-	[71]
14	CE-UV (2004)	Cranberry	95% ethanol:1.5 M HCl 85:15 *v*/*v*	Pn, Cy	-	[72]
15	CE-UV/Fluo (2007)	Grape skin and commercial extracts	0.8% HCl in ethanol–water mixture	Mv-3-glc, Pn-3-glc, Cn-3-glc, Dp-3-glc	-	[73]
16	CE-UV (2008)	Strawberries (cv. Camarosa)	acidified water (3% formic acid)	Pg-3-glu, Pg-3-rut and Cy-3-glc	-	[74]

### 4.4. Capillary Zone Electrophoresis (CZE) Using UV/Vis and Mass Spectrometry

Capillary Zone Electrophoresis (CZE), often referred to as Capillary Electrophoresis (CE), is a simple and convenient tool for analyzing a wide range of samples. The advantages include high resolution, short analysis time, small sample size, little solvent waste, high sample throughput, and relatively low cost. However, the method has limitations in structural analysis of separated peaks and becomes more powerful only if standards are used for quantitative or qualitative analysis. Other limitations include low reproducibility and sensitivity issues unlike LC systems. Among several separation modes, CZE and Micellar Electro Kinetic Chromatography (MEKC) are commonly used to separate small molecules or polyphenols.

Bednar et al. [66] showed that the selectivity of MEKC can be improved to allow the separation of six structurally close anthocyanins (Mv-3,5-diglc, Mv-3-glc, Mv-3-gal, Pg-3-glc, Cy-3,5-diglc, and Cy-3-gal) using a high content of borate. The optimized method was applied to a grape skin extract and found that Mv-3-glc was the main anthocyanin.

Similarly, Ichiyanagi et al. [67,68,69,70,71] (Table 3) published the separation of bilberry anthocyanins (or wild-type blueberry) analysis using borate-based buffer and studied their mobility behaviors. In these studies, the anthocyanin composition of different wild-type blueberry sources was evaluated and used for the kinetic studies of 12 anthocyanin reactivities towards ROS [67]. Twelve major peaks were separated and ten of them were identified by comparison with reference compounds of fifteen anthocyanins contained in bilberry extract. In addition, one of the remaining peaks was assigned as Dp-3-ara, based on a mobility comparison [68]. The acid hydrolysis of anthocyanins in TFA solution subjected to thermo-decomposition reaction at 95 °C which was determined primarily by the conjugated sugar type. The hydrolysis rate constants for the glycosides were in the following order, ara > gal > glc without the aglycon structure. However, four other minor anthocyanins (Mv-3-ara, Pn-3-gal, Pn-3-ara, and Pt-3-ara) in the bilberry extract [67,68] could not be identified because of the lack of authentic standards, although the possible migration time by electrophoretic rules in which both molecular weight and charges affect the mobility were discussed. Another study [69] discussed the structure–reactivity relationship for 12 anthocyanins found in bilberries towards 2,29-azobis(2-amidinopropane) dihydrochloride (AAPH) radicals, H_2_O_2_, and *t*-BuOOH. The reactivity towards peroxides was more remarkable than the reaction towards AAPH radicals. The reactivity of anthocyanins was mainly governed by the aglycon structure, and not by the type of sugar moiety. Ichiyanagi et al. further studied the reactivities of these anthocyanins towards other physiologically important reactive species, nitric oxide (NO) and peroxynitrite (ONOO^‒^) and also discussed the structure–reactivity relationships among them [70]. Lastly, the same author [71] purified these minor anthocyanins and determined their mobility behaviors. The mobility behavior of fifteen anthocyanins in bilberry was investigated by plotting the migration time against the molecular weight to numbers of a free phenolic group in the molecule ratios. Correlation between these variables was observed for a series of anthocyanins with the same conjugated sugar. Watson et al. [72] studied two aglycones (cyanidin and peonidin) in cranberries using acidic buffer at 525 nm and the results were compared with HPLC. Priego Capote et al. [73] developed a screening method for main phenolic compounds including Mv-3-glc, Pn-3-glc, Cy-3-glc, Dp-3-glc in residues of grape skin using a CE-UV/Fluorescence method. The analytes were separated using a 50 mM sodium tetraborate with 10% methanol (pH 8.4) solution as background electrolyte. Comandini et al. [74] assessed the CZE method for analyzing three main anthocyanins (Pg-3-glc, Pg-3-rut, Cn-3-glc) in strawberries at 510 nm. Acidic buffer solutions of pH < 2 were employed to maintain the stability in the flavylium cation form. The method was compared with HPLC, and both methods yielded similar results.

### 4.5. CE with MS Detection

Capillary electrophoresis (CE) has become prominent because of its ability to separate complex anthocyanin mixtures. The usefulness is in terms of composition, authenticity or adulteration, processing and quality of samples. Traditionally, the common detector coupled to CE instrumentation has been UV/Vis spectrophotometry, but the more sophisticated detection method of mass spectrometry is growing. CE-MS provides important advantages given the combination of the separation capabilities of CE and the power of MS as an identification, characterization and confirmation method. Bednář et al. [75] proved that the analysis of anthocyanins could be performed in both basic and acidic electrolytes and that sometimes borate buffer helps in separation of diastereomers. This technique was utilized to analyze polyphenolics in red grapes, wine and grape must samples. It can be a useful tool for wine evaluation and monitoring wine production.

The literature on anthocyanin analysis using HPTLC/TLC, CE, IR and NMR is in its infancy, with only a handful of articles published in the literature.

### 4.6. Liquid Chromatography Using UV and MS

High-performance liquid chromatography (HPLC) is the most widely used method for identifying and quantifying anthocyanins. In general, the analytical parameters used in the literature show uniform conditions for identifying anthocyanins. The most used column is C18, while mobile phase composition mainly corresponds to water, acetonitrile, and methanol with acid modifiers such as formic acid, acetic acid, phosphoric acid, or trifluoroacetic acid (TFA). The acid presence in the mobile phase ensures that anthocyanin compounds will be mobilized in their cationic flavylium form, which has been described as possessing its highest absorbance around 520 nm. During the HPLC analysis of anthocyanins and other compounds, retention times and peak areas can be strongly influenced by the column temperature, mobile phase composition, or the complexity of the matrix in which they are embedded. The detection of anthocyanins is often performed by Diode Array Detector (DAD) in mass-spectrometry (both MS and MS/MS) which is often coupled with an electrospray ionization source (ESI) [76]. These methodologies have shown satisfactory results for identifying and quantifying anthocyanins [15,40,77,78,79,80,81,82,83,84,85,86,87,88,89,90,91,92,93,94,95,96,97,98,99,100,101,102,103,104,105,106,107,108,109,110,111,112,113,114,115,116,117,118,119,120,121,122,123,124,125,126,127,128,129,130,131,132,133,134,135,136,137,138,139,140,141,142,143,144,145,146,147,148,149,150,151,152,153,154,155,156,157,158,159,160,161,162,163,164,165,166,167,168,169,170,171,172,173,174,175,176,177,178,179,180,181,182,183,184,185,186,187,188,189,190,191,192,193,194,195,196,197,198,199,200,201,202,203,204,205,206,207,208,209,210,211,212,213,214,215,216] (Table 4).

Nevertheless, Ultra-High-Performance Liquid Chromatography (UHPLC) provides better resolution, shorter elution times, and lower consumption of mobile phase than conventional HPLC methodologies. UHPLC also presents a high performance in the efficiency of the identification of peaks. The anthocyanin profile of diverse plant-related samples has been evaluated by HPLC-DAD. For example, in grapes, glucoside derivates of delphinidin, cyanidin, pelargonidin, peonidin, petunidin, and malvidin were identified [79]. Similarly, in grape skin samples, Pt-3-glc and Mv-3-glc were the major compounds [81]. HPLC-DAD can be also coupled to MS, which provides more accurate identification, since mass information is considered in the analysis and data processing. HPLC-DAD-MS has been employed with different matrixes, such as strawberries, where Cy-3-glc, Pg-3-glc, Pg-3-rut (tentative), Pg-3-suc-glc, and Pg-3-ara were identified [77]. Regarding HPLC coupled to MS, this approach has been employed to analyze the anthocyanin composition of different samples. For example, Cy-3-glc, Cy-3-rut, and Pg-3-glc have been identified in *E. edulis* extracts. In strawberries, glucoside derivates of cyanidin, delphinidin, pelargonidin, and malvidin were identified [78]. In muscadine grapes, 3,5-di-*O*-glucosides of cyanidin, delphinidin, and petunidin were identified as the major anthocyanins [80].

When the retention times of anthocyanins carrying the same sugar moiety were compared, it was seen that the aglycon structure determined the sequence of their retention times, therefore, the retention times were in the following order: delphinidin < cyanidin < petunidin < pelargonidin < peonidin < malvidin. On the other hand, when anthocyanins carrying the same aglycon structure were compared, the sequence of retention times was as follows: galactoside < glucoside < arabinoside [78]. It was also revealed that the hydrolysis rate of each anthocyanin was determined primarily by the type of conjugated sugar and not by the aglycon structure. The rate constant of anthocyanin hydrolysis was in the following order, arabinoside > galactoside > glucoside irrespective of aglycon structure [68]. In general, the most common option when using HPLC techniques is the selection of C18 columns and the modification of the mobile phase acidity by increasing the percentage of acid or by changing the type of acid.

Chromatography-based techniques were reported for quantifying anthocyanins in açai berries using liquid chromatography as a standalone method or hyphenated with mass spectrometry using various extraction solvents such as acidified organic solvents in 5–20% *v/v* of aqueous solution [84,88,91,93,95,96]. In 2008, Vera de Rosso et al. used an LC-MS/MS method to quantify the anthocyanins from açai berries based on PDA and to identify the anthocyanins based on their molecular mass and respective aglycone fragment ion [88]. Agawa et al. reported an HPLC-DAD method to quantify the anthocyanins in different part/s of açai berries, i.e., pulp, mesocarp, and endocarp. Among all these fruit parts, based on Dry Extract Weight, the pulp had the highest amount of anthocyanin, (34.1 mg/g DEW) followed by mesocarp (18.5 mg/g DEW), and the least amount of anthocyanins was found in endocarp (1.6 mg/g DEW). This study helps understand the distribution of polyphenols among various fruit parts. According to Dias et al., Cy-3-glc and Cy-3-rut contribute the majority of anthocyanin composition quantifying anthocyanins using an UHPLC-PDA method [95]. To study the antioxidant activity of açai berries [85,92,94], Poulose et al. studied the effect on signaling pathways linked to inflammation (microglial cells/neuroglial cells) in açai fruit pulp fractions which were prepared in a sequential manner using solvents from non-polar to polar solvents to prepare the fractions (hexane, chloroform, ethyl acetate, acetonitrile, ethanol and methanol solvents) [96].

Apples are well known for their nutritional values, good fiber content, vitamin C, and rich essential minerals. Around 50+ cultivar varieties of apples exist and most are used in food processing such as jams, pies, juices, and as flavoring agents. The studies reported for anthocyanin content using liquid chromatography/mass spectrometry of red apple peel were discussed. Compared to the flesh, the peel is an excellent source of anthocyanins with antioxidant activity [85,103]. In 2006, Sadilova et al. studied the anthocyanin content difference between peel, flesh, and their combination in red Weirouge apples [85]. This study provides insights into anthocyanin content in apple peel compared to their flesh and apple with peel. The majority of studies employed application of liquid chromatography to quantify anthocyanins at 520 nm absorption maxima [85,90,94,103]. In recent years, Oszmiański et al. studied phytochemical analysis of different apple varieties along with chemometric studies using LC-QToF-MS analysis data. HCA and PCA models were used to differentiate the apple varieties based on high-resolution mass data [103]. In addition, Oszmiański et al. performed antioxidant assays such as DPPH, ABTS, and FRAP for the various apple varieties.

Bilberries are native to Europe and contain diverse anthocyanins, i.e., cyanidin, delphinidin, malvidin, peonidin and petunidin glycosides. Various studies have been reported to study the content of anthocyanins using chromatographic techniques such as LC and TLC. In most cases, bilberry and blueberry extracts were used as reference materials for mixtures of diverse anthocyanins. In 2001, Dugo et al. reported a study for qualitative anthocyanin differentiation based on HPLC-PDA-MS in bilberry, blackberry, and mulberry extracts. This study concluded that bilberry extract contains 14 different anthocyanins [82]. In 2008, Riihinen et al. reported the difference in anthocyanin quantity between bilberry pulp and peel. This study concluded that there were 20 times higher amount of anthocyanins in the peel of bilberries compared to the pulp [87]. Besides antioxidant activity studies, bilberries were tested for acute cardioprotective and cardiotoxic effects. Ziberna et al. have extensively conducted their studies on bilberries for cardioprotective and toxic activities in ischemia-reperfusion injuries [92]. Benvenuti et al. have reported using HPLC for the analytical characterization of anthocyanins in bilberries along with their commercial products. Chemometrics analysis was reported for relative commercial products grouping based on their anthocyanin profiles [100].

Blackberries contain fewer anthocyanins than bilberries or blueberries [82]. Blackberries were studied for their phenolic compounds, ellagitannins, and flavonoids using reverse phase liquid chromatography standalone and coupled with mass spectrometry [82,83,86,89,97,98,99,101,102] and green chromatography approach using low toxicity solvents [132]. These HPLC-based methods were used to study the differences between different genotypes, to quantify anthocyanins in wild and cultivated varieties, and to study the authenticity of commercial preparations in comparison to the fresh berries or their extracts. Blackberry anthocyanin extracts were tested for their antioxidant, antibacterial and bioaccessability in the GI tract and colonic levels [83,89,97,99,102,132]. Cho et al. reported the separation of different flavonoid glycosides from various berries of different genotypes and their antioxidant capacities to evaluate the potential genotypes exhibiting antioxidant activity. Cy-3-glc and Cy-3-rut are the major anthocyanins reported from blackberries [83]. Jara-Palacios et al. reported a comparative anthocyanin study for red berry pomaces including the application of Stepwise Linear Discriminant Analysis (SDLA) chemometrics to differentiate red berries based on their chromatographic profiles. This method utilizes acidified (1N HCl) aqueous methanol as the extraction solvent for anthocyanin analysis [102].

Blackcurrant berries contain Dp-3-rut and Cy-3-rut as major anthocyanins based on reports involving isolation and identification of anthocyanin using an HPLC-DAD-MS method having antioxidant activities. Based on previous reports, blackcurrants contain 130–587 mg of anthocyanins per 100 g of fresh weight (FW) berries [105,107,113,118]. In addition to above mentioned major anthocyanins, blackcurrants have Dp-3-glc, Cy-3-glc and Pt-3-(6-*coumaroyl*) glc in minor amounts.

Blueberries are the most common berries used for human consumption in their raw form or processed foods such as juices, candies, jams, and as flavoring agents, etc. Blueberries contain diverse kinds of anthocyanins on a qualitative basis as well as on a quantitative basis. The anthocyanin content (42–6270 mg/100 g FW) varies depending on the sub-variety. Three varieties of blueberries are most commonly encountered; lowbush (*V. angutifolium*), highbush (*V. corymbosum*), and rabbit-eye (*V. ashei*). Blueberries contain cyanidin, malvidin, peonidin, petunidin, and delphinidin glycosides. The quantitative amount of anthocyanins based on 200 g Fresh Weight (FW) of berry varies depending upon the variety, i.e., lowbush blueberries contain 110–725 mg/100 g FW [106,109,110], highbush blueberries contain 42–6270 mg/100 g FW [108,112,115,116,118,120,129,138], and rabbit-eye blueberries contains 128–287 mg/100 g FW [108,137]. Various reports assess anthocyanin content qualitatively and quantitatively using liquid chromatography hyphenated with mass spectrometry. In 2000, Wang et al. developed HPLC and Matrix-Assisted Laser Desorption/Ionization Time of Flight Mass Spectroscopy (MALDI-ToF-MS) methods to quantify anthocyanins from blueberries. The study concludes that no substantial difference is observed in the amount of anthocyanins from blueberries via two different techniques [104]. In 2016, Li et al. reported an HPLC-PDA-MS method for chemical profiling of different blueberry varieties, i.e., Duke, Bluecrop, Northland, Northblue, and St. Cloud. This study utilized PCA-based chemometrics to differentiate the varieties by assessing LC-MS data [124]. In 2018, Sun et al. performed an organ-specific study to assess the content of anthocyanins in blueberries. The study reveals that quantities of anthocyanins are 40 times higher in fruit skin than in fruit pulp [129]. In 2021, Li et al. reported a study to assess the efficiency of extraction methods from conventional (solvent and ultrasonic extraction) to enzymatic extraction for anthocyanins and their antioxidant properties following each extraction method [134]. Several reports [83,97,102,104,106,108,109,110,111,112,114,115,116,117,118,119,120,121,122,123,124,125,126,127,128,129,130,131,132,133,134,135,136,137,138] observed that the quantity of anthocyanins in blueberries varies based on the place, collecting/harvesting season, extraction solvent, extraction method, and method of analysis.

The edible fruit of sweet cherries, a member of the Rosaceae family, contains substantial amounts of anthocyanins. Sweet cherries contain Cy-3-glc and Cy-3-rut as major anthocyanins. According to previously published reports, the quantity of anthocyanins varies from 0.3–642 mg/100 g FW [139,140,141,142,144,145,146]. The majority of reported methods used for quantification of anthocyanins/phenolic compounds in sweet cherries used HPLC at 520 nm [89,140,141,142,143]. In 2014, Crupi et al. reported metabolomic profiling of flavonoids from sweet cherries using LC-PDA-MS method to identify and quantify the anthocyanin composition [145]. In 2021, Hu et al. reported an LC-QToF-MS study to characterize the phenolic compounds from Australian-grown sweet cherries along with their potential antioxidant activities [181].

Chokeberries contain three cyanidin aglycone-based anthocyanins in major proportions, viz., Cy-3-ara, Cy-3-gal, and Cy-3-glc. Qualitative analysis of chokeberries [154], characterization of anthocyanins [154], anthocyanin content variation based on cultivars [155], and chemometric studies of chokeberry fruits [40] were reported in the literature. The content of anthocyanins varies from 249–737 mg/100 g FW based on different chokeberry cultivar samples. 0.1% acidified ethanol containing 20% aqueous solution was used for the qualitative analysis of chokeberry fruits. In addition, 6% formic acid in methanol was recently used as an extraction solvent for chemometric studies [40]. In 2020, Zielińska et al. reported a chemometric study based on ^1^H-NMR and LC-MS data to differentiate chokeberry varieties using principle component analysis (PCA) based chemometrics [40]. Chokeberry samples containing anthocyanins were reported for their antioxidant activity [154,155].

Cranberries have great commercial value because of their anthocyanin content and can be readily used in processed foods. Two different varieties of cranberries, large American cranberries (*V. macrocarpon*) and small European cranberries (*V. oxycoccus*), are generally used for human consumption. Both varieties of cranberries contain Cy-3-gal, Cy-3-glc, Pt-3-gal, Cy-3-ara, Pn-3-gal, and Pn-3-ara. In 2004, Seeram et al. tested cranberry extracts for inhibitory effects against Hep-G2 liver cancer cells. The study mainly focuses on the qualitative analysis of phytochemical constituents present in American cranberry extracts [188]. Further, Brown et al. developed and validated an HPLC-UV/Vis method to quantify the anthocyanins present in American cranberries [189]. Small European cranberries were majorly used as a food ingredient in processed foods of commercial value. The major anthocyanins in the small European cranberries are similar to that of large American cranberries. Optimization of ultra-sound assisted extraction of anthocyanins from cranberries was reported using 60% ethanol at a solid-to-liquid ratio of 1:30 g/mL. This reported study resulted in an unreported anthocyanin characterization, i.e., pelargonidin-3-(6-*malonyl*)-glucoside [190]. In 2009, Cesonienė et al. reported antibacterial activity for the European cranberry samples extracted using acidified (0.1N HCl *v/v*) ethanol containing 5% aqueous solution. European cranberries contain anthocyanins of 41–360 mg/100 g FW [106,162].

Crowberry is less familiar compared to other anthocyanin-containing berries. Crowberry is acidic in taste and is mostly used in wines, juices, and jellies. In 2010, Ali et al. reported a study that measures variations in anthocyanin concentration of wild crowberry populations using HPLC-DAD and HPLC-ESI-MS/MS methods [156]. The major anthocyanin varies from a sample collected in Finland to the northernmost and western regions. The sample collected from Finland shows Dp-3-gal as a major anthocyanin metabolite whereas the samples collected from the northernmost and western regions show Cy-3-gal as a major secondary metabolite. This significant variation in anthocyanins indicates the synthesis of anthocyanins in modified environmental conditions [156].

Elderberries have a high commercial value, mainly due to anthocyanins being present in high amounts. They are used in processed food such as jams, jellies, pies, ice creams, yogurts, juice, syrups, alcoholic beverages and dietary supplements. The most commonly used technique in analyzing elderberries is high-performance liquid chromatography coupled with UV or visible detection with or without mass spectrometry to mainly measure the anthocyanin and flavonol composition [147,163,191,192,193,194]. The European elderberry dry extract monograph of the United States Pharmacopeia includes a HPLC-Vis method for the anthocyanins with detection at 535 nm [195]. Mikulic-Petkovsek et al. detected 19 anthocyanins in four elderberry species and eight hybrids with content varying among these analyzed samples [163]. Cy-3-sam and Cy-3-glc were the most abundant anthocyanins in *S. nigra* fruits which agrees with the results reported by other authors mentioned above. In Europe, cultivars are mainly derived from *S. nigra* and, in the USA, most cultivars belong to *S. canadensis*. Similarly, Lee and Finn also found Cy-3-glc and Cy-3-sam as the major anthocyanins in *S. nigra*, while Cy-3-(*E*)-*p*-cou-sam-5-glc was found as the major pigment in *S. canadensis* [147]. Two elderberry cultivars were evaluated: ‘Korsør’ (Denmark) and ‘Haschberg’ (Austria). The total content of anthocyanins was 400 mg/100 g to 806 mg/100 g from 2004 to 2005 for ‘Korsør’. Cultivar ‘Haschberg’ accumulated from 391 mg/100 g to 657 mg/100 g from 2004 to 2005 of total anthocyanins. Veberic et al. investigated two cultivars (Haschberg and Rubini) and three selections (13, 14, 25) of black elderberry (*S. nigra*) [191]. The anthocyanin content and quercetin profiles of these samples were established using HPLC-DAD-MS. Five cyanidin-based anthocyanins were identified, and the most abundant anthocyanin in elderberry fruit was Cy-3-sam, which accounted for more than half of all anthocyanins identified in the berries. The ‘Rubini’ cultivar had the highest amount of the anthocyanins identified (1265 mg/100 g FW) and the lowest amount was measured in berries of ‘Selection 14’ (603 mg/100 g FW). The ‘Haschberg’ cultivar contained relatively low amounts of anthocyanins in ripe berries (737 mg/100 g FW). Kaack et al. studied the content and composition of anthocyanins in six elderberry cultivars [193]. They identified Cy-3-sam-5-glc, Cy-3,5-diglc, Cy-3-sam, Cy-3-glc, quercetin-3-rut, and quercetin-3-glc. The highest total content of anthocyanins in the fruits, 20.2 g/kg, was found in the ‘Finn Sam’ cultivar followed by the cultivars ‘Sampo’ (19.0 g/kg), ‘Haschberg’, ‘Mammut’ (17.7 g/kg), ‘Samocco’ (14.8 g/kg) and ‘Samdal’ (14.4 g/kg). In addition, Młynarczyk, K et al. showed that the elderberry fruit had higher phenolic and anthocyanin contents when growing in a well-organized orchard (4638.2 mg CGE/100 g DW) than in the wild (3071.0 mg CGE/100 g DW) [196]. Elderberries can therefore be considered a significant source of anthocyanins for food, medicinal, and other needs. Several authors have described that the largest group of polyphenols in elderberries were anthocyanins and anthocyanidins. The work based on coupling untargeted and targeted metabolomic approaches appears to be worthwhile [182] in order to better depict the anthocyanin composition of elderberry in an unbiased manner,

Goji berries belong to family Solanaceae, and the fruits are integral to traditional Chinese, Korean, and Japanese medicine. These berries contain a wide variety of anthocyanins, most of which are acylated glycosides of cyanidin, delphinidin, petunidin, peonidin and malvidin. Most reported studies use acidified organic solvents such as ethanol/methanol as an extraction solvent. Kosar et al. reported the anthocyanin composition and antioxidant activity of goji berry samples using non-polar petroleum ether followed by ethyl acetate and methanol (increase in polarity). The reported amount of anthocyanins vary from trace to 3.8 mg/g of dry extract weight (DEW). The study includes antioxidant activity for the different solvent extracts using the DPPH assay [148]. Zheng et al. reported the composition of anthocyanins from Qinghai-Tibet Plateau as being acylated based on their elution time and respective aglycone and acyl moiety molecular ions [149]. In 2018, Sang et al. reported the quantification of anthocyanins from goji berries using Deep Eutectic Solvents (DES) as an extraction approach with choline chloride and 1,2-propanediol in different molar ratios. In addition, a 2D-LC method coupled with mass spectrometry was used to characterize the anthocyanins along with their quantitative profiles [184]. Recently Cheng et al. reported fingerprinting of anthocyanin profiles from goji berries for quality assessment and geographical origin identification. Their study uses a PCA based chemometrics approach to differentiate the goji berries based on their quality and geographical distribution [183].

Red grapes are common fruits used for human consumption and commercial purposes. The skin of red grapes contains a variety of acylated anthocyanins. The quantity of anthocyanins varies from 2.4 to 790 mg/100 g FW of grape skin. Cho et al. reported the anthocyanin content variation in the genotypes such as A-1575, A-2467, A-2663, Cynthiana, and Cabernet Sauvignon from 38 to 790 mg/100 g FW of grape skin. The flavonoid glycosides were qualitatively characterized using an HPLC-PDA-MS method [83]. Differences in the extraction solvent and type of grape peel generate variation in the anthocyanin’s composition. Anthocyanins from grape peel based on their genotypic variety were studied, and changes in the phenolic contents during ripening stage show the metabolomic changes in PCA-based chemometrics [165]. Various HPLC-PDA and HPLC-PDA-MS reports show that the grape skin contains acylated anthocyanins, which are high in dry extracts, i.e., 700–56,470 mg/100 g DEW [79,157,165,185,186,187]. Košir et al. reported anthocyanin identification using NMR and LC-MS techniques using HCA and Regularized Discriminant Analysis (RDA) to differentiate the anthocyanins [197].

Mulberries belong to the Moraceae family, and the fruit color is based on anthocyanin composition. These berries show Cy-3-sop, Cy-3-glc, Cy-3-rut, Pg-3-glc and Pg-3-rut as major anthocyanins. Dimitrijević et al. reported the comparative view of *Morus sp.* using HPLC-PDA analysis using acetone containing 1% HCl as the extraction solvent. The quantity of anthocyanins present in black mulberries is 125 mg/100 g DEW [146]. Optimization of microwave extraction for the mulberry anthocyanins using acidified methanol concentrations varying from 10 to 70% containing 1% TFA was reported followed by qualitative analysis [164]. In 2017, Sang et al. reported a green chromatography approach for the anthocyanin determination in berries using an ethanol and α-hydroxy acid aqueous solution as the mobile phase with a reverse phase C_18_ column. This approach provides better insights for applying less toxic organic solvents for anthocyanins separation solvents [98].

Pomegranate peel is a rich source of anthocyanins which are well known for their antioxidant activity [175,176,177,179,180]. The amount of anthocyanin in pomegranate peel varies from trace to 344 mg/100 g of FW. This peel contains diglucosides of delphinidin, cyanidin, and pelargonidin. Reports indicate extraction of anthocyanins from fresh/dried peel using acidified methanol as the extraction solvent. Apart from its antioxidant activity, pomegranate peel extract was tested for antiproliferative activity and cytotoxic assays [176]. Brighenti et al. [178] reported metabolomic fingerprinting of pomegranate peel anthocyanins/polyphenols using acidified methanol containing 20% aqueous solution. PCA-based chemometrics were used in this study to differentiate the polyphenols from different varieties of pomegranate peel. Balli et al. reported the characterization of anthocyanins from pomegranate peel and juice separately. The study shows 61 mg/100 mL of juice and 68 mg/100 mg FW of peel. In addition, the study reported antioxidant activity, α-amylase activity, and tyrosine inhibitor activities for both juice and peel portions from pomegranate [180].

Two varieties of raspberries are used for human consumption as well as in processed food preparations, i.e., red raspberries (*R. idaeus*) and black raspberries (*R. occidentalis*). Anthocyanins from red raspberries showed Cy-3-sop and cy-3-glc in major amounts. Pelargonidin acylated derivatives and glycosides were reported in the literature from red raspberries. The quantity of anthocyanins varies from 0.1 to 134 mg/100 g FW, 76 to 365 mg/100 g DW, and 336 to 1030 mg/100 g DEW [15,89,97,102,117,132,152,153,158,159,160,161,169,170] of red raspberry variety. Reported studies indicate usage of acidified (HCl or formic acid) methanol alone or ethyl acetate extraction followed by acidified methanolic extraction for the qualitative/quantitative analysis of anthocyanins is suitable for the red variety. Chemometric studies such as SDLA and PCA were used for the comparative analysis of red berries and the physicochemical characterization of red raspberries [102,160]. Mullen et al. characterized the anthocyanins in red raspberry fruit using an LC-MS approach and tested acidified methanolic extracts for antioxidant and vasorelaxation activities [170]. In recent years, red raspberries were studied to quantify anthocyanins using an aqueous ethanolic mixture acidified with citric acid/HCl and reported their antioxidant and antibacterial effects against *H. pylori* infection [132,160,161]. The anthocyanin content in black raspberries varies from 2885 to 11,109 mg/100 g DEW [132,150]. Black raspberries contain Cy-3-sam, Cy-3-xyl-rut, and Cy-3-rut in major proportions [15,132].

Strawberries are a rich source of anthocyanins, and their qualitative and quantitative analyses were reported using liquid chromatography/mass spectrometry techniques. Strawberries contains mainly Cy-3-glc, Pg-3-rut, and Pg-3-glc. The quantity of anthocyanins varies from 0.4 to 84 mg/100 g FW, 97 to 759 mg/100 g DW, and 520 mg/100 g DEW of berries [77,97,136,151,152,153,166,167,168,171,172,173,174]. Identification and characterization of phenolic compounds from strawberries utilized HPLC-PDA and HPLC-PDA-MS methods [77,97,136,151,152,153,166,167,168,171,172,173,174]. Fernández-Lara et al. reported an assessment of phenolic profile differences in various strawberry cultivars using a Forward Linear Discriminant Analysis (FLDA) chemometric approach [171].

**Table 4 molecules-28-00560-t004:** Distribution, extraction method, % yield, detection methods, analysis purpose and references of anthocyanins in selected fruits and berries.

Açai (*E. oleracea*)											
#	Source (Year)	Anthocyanins	Extraction Solvent	% Yield	Conditions		Detection Method	Purpose of Analysis	Chemometrics	Pharmacological Activity	[Ref]
					Stationary Phase	Mobile Phase					
1	Açai berries (2005)	Cy-3-glc; Cy-3-rut; Pn-3-rut;	Millipore water	13–463 mg/L	HPLC-Vis Max-RP 80A column (150 × 4.6 mm, 4 µm)	Gradient program	HPLC-MS and HPLC-UV 525 nm	Antioxidant capacities of açai fruits	-	Antioxidant activity	[84]
2	Açai berries (2008)	Cy-3,5-hex-pent; Cy-3-glc; Cy-3-rut; Pg-3-glc; Pn-3-glc; Pn-3-rut; Cy-3-Ac-hex;	95% ethanol/1.5 N HCl	282.5–303.7 mg/100 g FW	Shim-pack CLC-ODS column (250 × 4.6 mm, 5 µm)	Gradient program	HPLC-PDA-MS/MS 520 nm	Determination of anthocyanins	-	-	[88]
3	Açai fruit pulp (2009)	Cy-3-glc; Cy-3-rut; Pn-3-rut;	Aqueous solution adjusted to pH to 3.5 with citric acid	224.7 mg/100 g FW	Synergi Hydro-RP 80 OA (150 × 2 mm, 4 µm)	Gradient program	HPLC-PDA-ESI-MSn	Phytochemical composition and thermal stability of açai species	-	Antioxidant activity	[91]
4	Açai fruit (pulp, mesocarp, and Endocarp) (2011)	Cy-3-glc; Cy-3-rut;	80% ethanol containing 0.5% acetic acid	Pulp: 34.1 mg/g DEW; Mesocarp: 18.5 mg/g DEW; Endocarp: 1.64 mg/g DEW	Capcell Pak ACR (250 × 4.6 mm, 5 µm)	Gradient program	HPLC-DAD	Anthocyanins in different parts of açai fruit	-	Antioxidant activity	[93]
5	Açai berries pulp (2012)	Cy-3-glc; Cy-3-rut; Dp-3-glc; Mv-3-glc; Pg-3-glc; Pn-3-glc;	400 mL of hexane followed by 400 mL of chloroform for 36 h. The leftover pulp extracted with ethyl acetate for 36 h at room temp.	EtOAc fraction: 0.12 μg/mg DEW Acetonitrile fraction: 2.3 μg/mg DEW Methanol fraction; 11.3 μg/mg DEW Ethanol fraction: 8.9 μg/mg DEW	Inertsil ODS-2 (250 × 4.6 mm, 5 µm)	Gradient program	HPLC-PDA 520 nm	Attenuation of inflammatory stress signaling in mouse brain BV-2 microglial cells	-	Effect on signaling pathways linked to inflammation in microglial cells/Neuroinflammation	[96]
6	Açai fruit (2012)	Cy-3,5-diglc, Cy-3-glc, Cy-3-rut, Pg-3-glc, Pn-3-glc, Pn-3-rut	Methanol (0.1% HCl)	Major anthocyanins (Cy-glc and Cy-rut) 489–584 mg/kg FW	HSS C18 (100 × 2.1 mm, 1.8 µm)	Gradient program	UHPLC-PDA	Validated UHPLC-PDA method for anthocyanin quantification	-	-	[95]
Apples-Red peel (*M. domestica*)											
7	Red apples Peel and Whole (2006)	Cy-3-gal; Cy-3-glc; Cy-pent-rha; Cy-3-arab; Pn-3-gal; Cy-3-maloyl-gal; 5-carboxy-pyrano-Cy-hex; Cy-7-arab; Cy-3-xyl; Cy-3-pent;	Aq. acetone (pH 1.0 water acidified with TFA/acetone 30/70 *v*/*v*)	Apple with peel: 10.8 mg/100 g FW; Apple flesh: 8.1 mg/100 g FW; Apple peel: 20.6 mg/100 g FW;	Sunfire C18 (250 × 4.6 mm, 5 µm)	Gradient program	HPLC-DAD-MS 520 nm	Anthocyanins pattern in red fleshed Weirouge apples	-	Antioxidant property	[85]
8	Fuji apples Peel (2009)	Cy-3-gal; Cy-3-glc; Cy-3-arab; Cy-pent;	Methanol containing 1% *v*/*v* HCl and 1% *w*/*v* BHT	Normal: 11–24.8 mg/100 g FW; Hailnet: 18.2–26.1 mg/100 g FW	Gemini C18 (150 × 4.6 mm, 3 µm)	Gradient program	HPLC-PDA-MS 520 nm	Influence of light exposure on phenolic content	-	-	[90]
9	Red apples Peel (2012)	Cy-hex; Mv-pent; Cy-pentoxide; Cy-pent;	-	-	Suplelco ODS Hypersil (150 × 2.1 mm, 5 µm)	Gradient program	HPLC-DAD-MS/MS 530 nm	Investigation of phyto components in red apples by LC-MS and GC-MS	-	-	[94]
10	Apples Whole (2020)	Cy-3-gal;	10 mL of mixture containing methanol (30 mL/100 mL), ascorbic acid (2 g/100 mL) and acetic acid (1 mL/100 mL)	Cy-3-gal; 0–41 mg/100 g DW	Acquity BEH C18 (100 × 2.1, 1.7 µm)	Gradient program	LC-QToF-MS	Phytochemical analysis of old apple varieties	Hierarchical cluster analysis and Principle component analysis (PCA)	Antioxidative activity	[103]
Bilberry (*V. myrtillus*)											
11	Bilberry (2001)	Dp-glycosides; Cy-glycosides; Pt-glycosides; Mv-glycosides;	Aliquot of 5 mL of extract from fresh berries	-	Restek Pinnacle ODS (250 × 4.6 mm, 5 μm)	Gradient program	HPLC-ESI-MS	Identification of anthocyanins in different berries	-	-	[82]
12	Bilberry (2003)	Dp-glycosides; Cy-glycosides; Pt-glycosides; Mv-glycosides;	acetonitrile/TFA/water (49.5:0.5:50 *v*/*v*/*v*)	472.3 mg/100 g FW	Zorbax SB-C18 (150 × 4.6 mm, 5 μm)	Gradient program	HPLC-DAD 520 nm HPLC-DAD-MS	Isolation and identification of anthocyanins from berries	-	Antioxidant activity	[107]
13	Bilberries (2008)	Dp-glycosides; Cy-glycosides; Pt-glycosides; Mv-glycosides; Pn-glycosides;	10%A and 90%B solvents. Solvent A: acetonitrile/methanol (85/15 *v*/*v*) Solvent B: 8.5% aq. Formic acid	350–525 mg/100 g FW (Quant based on Aglycones content)	Phenomenex gemini C18 (150 × 4.6 mm, 5 μm)	Gradient program	HPLC-DAD 520 nm	Anthocyanins analysis in wild bilberries	-	-	[204]
14	Bilberries (2008)	Dp-glycosides; Cy-glycosides; Pt-glycosides; Mv-glycosides; Pn-glycosides;	-	Peel: 2025.6 mg/100 g Pulp: 104 mg/g FW	Lichrocart Purospher RP-18e (125 × 3 mm, 5 μm)	Gradient program	HPLC-DAD 520 nm	Organ specific distribution of phenolics in bilberries	-	-	[87]
15	Bilberries (2010)	Dp-glycosides; Cy-glycosides; Pt-glycosides; Mv-glycosides; Pn-glycosides;	SEP-PAK C18 cartridge. The cartridge was previously conditioned with 3 mL methanol and 5 mL 5 mM H_2_SO_4_ and washed with 5 mL of 5 mM H_2_SO_4_ and dried with nitrogen before eluted with 4 mL of methanol.	967.8 mg/100 g FW	Phenomenex gemini C18 (150 × 4.6 mm, 5 μm)	Gradient program	HPLC-DAD 520 nm	Pharmacological effects of bilberry anthocyanins	-	Antioxidant property, Acute Cardioprotective and Cardiotoxic effects	[92]
16	Bilberries (2012)	Dp-glycosides; Cy-glycosides; Pt-glycosides; Mv-glycosides; Pn-glycosides;	acetonitrile/water/formic acid (87:3:10 *v*/*v*)	6102–7465 mg/100 g DW	Phenomenex Luna C18(2) (250 × 4.6 mm, 3.0 µm)	Gradient program	HPLC-UV/Vis 520 nm	Analysis of anthocyanins in bilberries and their commercial products	-	-	[120]
17	Bilberries (2018)	Dp-glycosides; Cy-glycosides; Pt-glycosides; Mv-glycosides; Pn-glycosides;	2% HCl and methanol (5:95 *v*/*v*)	582–795 mg/100 g FW	Zorbax SB-C18 (150 × 4.6 mm, 5 μm)	Gradient program	HPLC-DAD-MS 520 nm	HPLC analysis of anthocyanins in bilberries and their food products	Principle component analysis (PCA)	-	[100]
Blackberry (*R. fruiticosus*)											
18	Blackberry (*Rubus sp.*) (2001)	Cy-glycosides; Pg-glycosides; Mv-glycosides;	Aliquot of 5 mL of extract from fresh berries was taken and centrifuged	-	Restek Pinnacle ODS (250 × 4.6 mm, 5 μm)	Gradient program	HPLC-ESI-MS	Identification of anthocyanins in different berries	-	-	[82]
19	Blackberry genotypes (2004)	Cy-3-glc; Cy-3-rut; Cy-3-xyl; Cy-3-malonyl-glc; Cy-3-dioxaloyl-glc;	methanol/water/FA (60:37:3 *v*/*v*)	Blackberry: 1.1–2.4 g/kg FW	Symmetry C18 (250 × 4.6 mm, 5 μm)	Gradient program	HPLC-PDA-MS PDA at 520 nm	Flavonoid glycosides estimation in various genotypes of red wine grapes, blackberry and blueberries	-	Antioxidant properties	[83]
20	Blackberries (*Rubus sp.*) (2007)	Cy-3-glc; Cy-3-rut; Cy-3-malonyl-glc;	70% aq. Acetone containing 2% formic acid	720–1010 mg/100 g DW	Lichrospher ODS-2 (250 × 4.6 mm, 5 μm)	Gradient program	HPLC-DAD-MS 520 nm	Analysis of phenolic compounds in blackberry sp.	-	-	[86]
21	Blackberries (2009)	Cy-3-glc;	after ethyl acetate extraction sample was acidified with 1 mL of 2 M HCl and extracted with 5 mL of methanol for 4–8 times.	97.7 mg/100 g FW	Omnispher C18 (250 × 4.6 mm, 5 μm)	Gradient program	HPLC-PDA 520 nm	Phenolic composition of berries	-	Antioxidant activity	[89]
22	Blackberries (2011)	Cy-3-glc; Cy-3-rut; Cy-3-xyl; Cy-3-(6-malonyl)-glc; Cy-3-(6-(3-OH-3-methylglutaroyl)-glc)	0.5% TFA acidified methanol.	323.3 mg/100 g FW	Hypersil ODS (200 × 4.6 mm, 5 μm)	Gradient program	HPLC-DAD 520 nm	Identification of anthocyanins from wild and cultivated blueberries	-	-	[202]
23	Blackberries (2016)	Dp-3-glc; Cy-3-glc; Cy-3-rut; Cy-3-xyl;	-	647 mg/100 g FW	Lichrocart RP-18 (250 × 4.6 mm, 5 μm)	Gradient program	HPLC-DAD 520 nm	Phenolic composition and antioxidant activity of different berries	-	Antioxidant activity	[97]
24	Blackberries (2017)	Cy-3-glc; Cy-3-xyl; Cy-3-malonyl-glc; Cy-3-dioxaloyl-glc;	1% TFA in ethanol	27230 mg/100 g DEW	TSK-GEL ODS-100 V (150 × 4.6 mm, 5 μm)	Gradient program	HPLC-DAD-MS 520 nm	Anthocyanin content in blackberries	-	-	[99]
25	Blackberries (2017)	Cy-3-gal; Cy-3-glc; Cy-3-rut; Cy-3-arab; Cy-3-xyl; Cy-3-malonyl-glc; Cy-3-dioxaloyl-glc;	0.1% HCl in 15 mL of ethanol	0.17 mg/mL	Zorbax Eclipse plus C18 (100 × 4.5 mm, 3.5 μm)	Ethanol and α-hydroxy acid aqueous solution	HPLC-DAD 520 nm HPLC-DAD-MS	Green chromatography for anthocyanins determination in berries	-	-	[98]
26	Blackberries (2018)	Cy-3-glc; Cy-3-xyl; Cy-3-(6-malonyl)-glc; Cy-3-(6-(3-OH-3-methylglutaroyl)-glc;	80:20 *v*/*v* methanol and water with 0.5% acetic acid	222 mg/100 g FW	Phenomenex gemini C18 (250 × 4.6 mm, 5 μm)	Gradient program	HPLC-DAD 520 nm	Anthocyanins and ellagitannins from blackberries	-	Bioaccessability studies in GI tract and colonic levels	[101]
27	Blackberries (2018)	Cy-3-glc; Cy-3-rut;	1 N HCl in 75% methanol	192.4 mg/100 g DW	Zorbax SB-C18 (250 × 4.6 mm, 5 µm)	Gradient program	HPLC-PDA-MS 520 nm	Comparative study of red berry pomaces	stepwise linear discriminant analysis (SLDA)	Antioxidant activity	[102]
28	Blackberries (2021)	Cy-3-glc; Cy-3-rut; Cy-3-xylosyl-rut;	80:20 ethanol/water	847–3465 mg/100 g DEW	-	-	HPLC-MS	Antibacterial effects of berries against *H. pylori* infection	-	Antibacterial activity	[132]
Blackcurrant (*R. nigrum*)											
29	Blackcurrant (2003)	Dp-3-glc; Dp-3-rut; Cy-3-glc; Cy-3-rut; Pt-3-rut	acetonitrile/TFA/water (49.5:0.5:50 *v*/*v*)	213.7 mg/100 g FW	Zorbax SB-C18 (150 × 4.6 mm, 5 μm)	Gradient program	HPLC-DAD 520 nm HPLC-DAD-MS	Isolation and identification of anthocyanins from berries	-	Antioxidant activity	[107]
30	Black currant (2011)	Dp-3-glc; Dp-3-rut; Cy-3-glc; Cy-3-rut; Pt-3-(6-coumaroyl)-glc;	5 g of frozen material extracted with 10 mL of extraction solvent., i.e., acetone/acetic acid (99:1)	162.8–180.4 mg/100 g FW	Zorbax SB C18 (150 × 4.6 mm, 5 μm)	Gradient program	HPLC-DAD-MS 520 nm	Characterization and quantification of phenolic compounds in blueberries, black and red currants		-	[118]
Blueberries (*V. corymbosum*)											
31	Blueberries (2000)	Pg-3-glc; Cy-3-glc; Pn-3-glc; Mv-3-glc	acetone/methanol/water/FA (40:40:20:0.1 *v*/*v*)	HPLC: 0.08–0.64 mg/mL MALDI-ToF-MS: 0.08–0.8 mg/mL	Zorbax SB-C18 (150 × 4.6 mm, 5 μm)	Gradient program	HPLC-PDA 520 nm MALDI-ToF-MS	Comparison of HPLC and MALDI-ToF-MS for anthocyanins analysis in blueberries	-	-	[104]
32	Blueberries (2001)	Dp-glycosides; Cy-glycosides; Pt-glycosides; Mv-glycosides; Pn-glycosides;	Acetone/water/acetic acid (70:29.5:0.5 *v*/*v*)	Lowbush blueberries: 1.7 mg/g FW Highbush blueberries: 2.2–2.8 mg/g FW	Zorbax C18 (150 × 4.6 mm, 5 μm)	Gradient program	HPLC-MS/MS	Identification of anthocyanins and procyanidins in blueberries and cranberries	-	-	[106]
33	Blueberry genotypes (2004)	Dp-glycosides; Cy-glycosides; Pt-glycosides; Mv-glycosides; Pn-glycosides;	methanol/water/FA (60:37:3 *v*/*v*)	Blueberries: 1.43–8.2 g/kg FW	Symmetry C18 (250 × 4.6 mm, 5 μm)	Gradient program	HPLC-PDA-MS PDA at 520 nm	Flavonoid glycosides estimation in various genotypes of red wine grapes, blackberry and blueberries	-	Antioxidant properties	[83]
34	Lowbush blueberry (*V. angustifolium*) (2006)	-	0.1% HCl in methanol	*V. angustifolium*: 350–725 mg/100 g FW	Zorbax C18 (150 × 4.6 mm, 5 µm)	Gradient program	HPLC-DAD 517 nm, 520 nm, 525 nm, 530 nm	Anthocyanin content in different vaccinium types	-	-	[109]
35	Wild blueberries (2007)	-	Ethanol at 77 °C, 26 °C, or 79 °C without acid (pH 5.4) or acidified with hydrochloric (pH 4.1), citric (pH 4.9), tartaric (pH 5.0), lactic (pH 4.8), or phosphoric acid (pH 4.6; 0.02% *v*/*v*)	24 mg/g DEW	Supelco C18 (250 × 4.6 mm, 4 µm)	Gradient program	HPLC-PDA 520 nm	Anthocyanins in wild blueberries	-	-	[110]
36	Blueberries (2007)	Dp-glycosides; Cy-glycosides; Pt-glycosides; Mv-glycosides; Pn-glycosides;	1 N HCl acidified methanol (85:15 *v*/*v*)—pH to 1.0	558.3 mg/100 g DW	Luna C18 (150 × 3 mm, 3 µm)	Gradient program	HPLC-PDA 520 nm and UPLC-ESI-MS	Anthocyanin content determination in berries	-	-	[153]
37	Blueberries (2008)	Dp-glycosides; Pt-glycosides; Mv-glycosides;	methanol/acetic acid/water (25:1:24 *v*/*v*)	-	Zorbax Eclipse XDB C18 (150 × 4.6 mm, 5 µm)	Gradient program	HPLC-ESI-MS 520 nm	Identification of scavenging compounds in blueberry extract by HPLC coupled to online ABTS based assay	-	Antioxidant activity	[111]
38	Highbush blueberries (*V. corymbosum*) (2008)	Dp-glycosides; Cy-glycosides; Pt-glycosides; Mv-glycosides; Pn-glycosides;	MeOH/Water/Acetic acid (25:24:1 *v*/*v*)	5.8–9.6 g/kg DW	Xterra MS C18 (150 × 2.1 mm)	Gradient program	LC-MS (SQD) PDA at 520 nm	Determination of anthocyanins content in various cultivars	-	-	[112]
39	Blueberries (*V. corymbosum*) (2009)	List of aglycones provided	0.1% HCl in methanol	Blueberry fruit skin: 6.2 mg/g FW; Blueberry pulp: 0.02 mg/g FW	Spherisorb ODS C18 (250 × 4.6 mm, 5 µm)	Gradient program	HPLC-DAD	Degradation of anthocyanins and anthocyanidins in blueberry jams/stuffed fish	-	-	[115]
40	Blueberries (2009)	Dp-glycosides; Cy-glycosides; Pt-glycosides; Mv-glycosides; Pn-glycosides;	MeOH: Water: TFA (70:30:1 *v*/*v*)	-	Gemini C18 (100 × 2.0 mm, 3 μm)	Gradient program	LC-MS-IT-ToF-MS(PDA 520 nm)	Characterization/Identification of anthocyanins	-	-	[114]
41	Highbush blueberries (*V. corymbosum*) (2009)	Dp-glycosides; Cy-glycosides; Pt-glycosides; Mv-glycosides; Pn-glycosides;	methanol/acetone/water/acetic acid (30:30:35:0.1 *v*/*v*)	Blueberries: 78.5 mg/100 g FW	Phenomenex Luna C18 (250 × 4.6 mm, 5 µm)	Gradient program	HPLC-PDA 520 nm	Effect of processing and storage conditions on phenolic compounds	-	Antioxidant activity	[116]
42	lowbush blueberries (*V. angustifolium*) (2010)	Dp-glycosides; Cy-glycosides; Pt-glycosides; Mv-glycosides; Pn-glycosides;	methanol with 0.1% HCl	Qualitative	Synergi RP-Max (250 × 2 mm, 4 µm)	Gradient program	HPLC-DAD-MS/MS	Anthocyanins analysis in berries	-	-	[117]
43	Different cultivars of blueberries (2011)	Dp-3-gal; Dp-3-glc; Dp-3-arab; Pt-3-gal; Pt-3-glc	methanol/acetic acid/water (25:1:24 *v*/*v*)	-	Zorbax Eclipse XDB C18 (150 × 4.6 mm, 5 µm)	Gradient program	HPLC-ESI/MS and online HPLC-ABTS 520 nm	Evaluation of antioxidant activity in Blueberry cultivars by HPLC-ESI/MS and HPLC-ABTS system	-	Antioxidant activity	[119]
44	Blueberry (*V. corymbosum*), (2011)	Dp-glycosides; Cy-glycosides; Pt-glycosides; Mv-glycosides; Pn-glycosides;	acetone/acetic acid (99:1 *v*/*v*)	42–83.6 mg/100g FW	Zorbax SB C18 (150 × 4.6 mm, 5 μm)	Gradient program	HPLC-DAD-MS 520 nm	Characterization and quantification of phenolic compounds in blueberries, black and red currants	-	-	[118]
45	Highbush blueberries (*V. corymbosum*) (2012)	Dp-glycosides; Cy-glycosides; Pt-glycosides; Mv-glycosides; Pn-glycosides;	acetonitrile/water/formic acid (87:3:10 *v*/*v*)	Highbush blueberries: 1.6–2.8 g/100 g DW	Phenomenex Luna C18(2) (250 × 4.6 mm, 3.0 µm)	Gradient program	HPLC-UV/Vis 520 nm	Analysis of anthocyanins in bilberries and blueberries	-	-	[120]
46	Blueberries (2016)	Dp-3-glc; Cy-3-glc; Pt-3-glc; Pn-3-glc; Mv-3-glc;	-	77.5 mg/100 g FW	Lichrocart RP-18 (250 × 4.6 mm, 5 μm)	Gradient program	HPLC-DAD 520 nm	Phenolic composition and antioxidant activity of different berries	-	Antioxidant activity	[97]
47	Blueberries of different varieties (2016)	Dp-glycosides; Cy-glycosides; Pt-glycosides; Mv-glycosides; Pn-glycosides;	methanol with 0.1% HCl *v*/*v*	Total Anthocyanidins: 108.1–300.6 mg/100 g FW	Zorbax Eclipse XDB C18 (250 × 4.6 mm, 5 μm)	Gradient program	HPLC-PDA-MS	Chemical profiling of different anthocyanins using HPLC-PDA-MS	PCA	-	[124]
48	Fresh skin of blueberry (2016)	-	0.1 M HCl solution 90:10 (*v*/*v*)	-	Analytical C18	Gradient program	HPLC-PDA 525 nm	Mobile phase variation studies using fruit and rose extract using HPLC method	-	-	[123]
49	Blueberries (2016)	Dp-3-glc; Mv-3-gal; Cy-3-gal; Dp-3-gal; Cy-3-rut; Mv-3-glc	Microwave assisted extraction: methanol/chloroform (2:1) Irradiation time: 20 min at 40 OC.	Hydroalcoholic extract: 9.9–45.5 µg/g DEW Organic extract: nd–31.65 µg/g DEW	GraceSmart RP 18 (250 × 4.6 mm, 5 µm)	Gradient program	HPLC-PDA 520 nm	Microwave assisted extraction on Italian blueberry varieties	-	Carbonic anhydrase inhibition	[125]
50	Blueberry wine lees (2017)	Dp-3-glc; Pt-3-glc; Mv-3-gal; Mv-3-glc; Mv-3-arab; Mv-3-Ac-gal; Mv-3-Ac-glc;	0.1% HCl acidified 70% (*v*/*v*) ethanol	-	TSK Gel ODS-100 Z (150 × 4.6 mm, 5 µm)	Gradient program	LC-MS DAD at 520 nm	Identification of anthocyanins from Blueberry wine lees	-	-	[127]
51	Bluehaven Highbush blueberries (2017)	Dp-glycosides; Cy-glycosides; Pt-glycosides; Mv-glycosides; Pn-glycosides; Acylated anthocyanins	MeOH with 1.3%FA (*v*/*v*)	-	Kinetex PFP (150 × 4.6 mm, 2.6 μm)	Gradient program	PDA at 520 nm	Development of fast chromatography with unique separation using PFP stationary phase	-	-	[126]
52	Highbush blueberries (*V. corymbosum*) (2018)	Dp-glycosides; Cy-glycosides; Mv-glycosides; Pn-glycosides;	-	Blueberry fruit skin: 672 mg/g FW; Blueberry pulp: 18.1 mg/g FW	Waters C18 column (250 × 4.6 mm, 5 µm)	Gradient program	HPLC-PDA 520 nm	Comparison of phytochemical profiles of highbush blueberries at different developmental stages	-	Antioxidant activity	[129]
53	Blueberry anthocyanin extract (2018)	Dp-3-glc; Cy-3-glc; Mv-3-glc; Pg-3-glc; Cy-3-arab; Pn-3-glc;	60% ethanol	-	Innoval ODS-2 C18 (250 × 4.6 mm, 5 µm)	Gradient program	HPLC-DAD-MS	Antioxidant activity evaluation of anthocyanin extracts and their protective effect against acrylamide induced toxicity in HepG2 cells	-	Antioxidant properties	[128]
54	Blueberries (2018)	Dp-glycosides; Cy-glycosides; Pt-glycosides; Mv-glycosides; Pn-glycosides;	1 N HCl in 75% methanol	1188.3 mg/100 g DW	Zorbax SB-C18 (250 × 4.6 mm, 5 µm)	Gradient program	HPLC-PDA-MS 520 nm	Comparative study of red berry pomaces	stepwise linear discriminant analysis (SLDA)	Antioxidant activity	[102]
55	Ripened blueberries (2021)	Dp-3-gal; Dp-3-glc; Cy-3-gal; Cy-3-glc; Cy-3-ara; Pt-3-ara; Mv-3-gal; Mv-3-glc; Mv-3-ara;	60% EtOH at 1:15 (Solid to liquid) ratio	-	Waters C18 column (250 × 4.6 mm, 5 μm)	Gradient program	HPLC-PDA-MS PDA at 520 nm	Antioxidant and bioaccessability of blueberry anthocyanins under in vitro simulated digestion	-	Antioxidant activity	[133]
56	Dried blueberries (2021)	Dp-glycosides; Cy-glycosides; Pt-glycosides; Mv-glycosides; Pn-glycosides;	Water as solvent (25 g/2 L)	147.59 mg/L	Zorbax C18 (250 × 4.6 mm, 5 μm)	Gradient program	PDA at 520 nm	Optimization of membrane filtrations for aqueous blueberry extracts	-	-	[131]
57	Blueberries (2021)	Dp-glycosides; Pt-glycosides; Mv-glycosides; Pn-glycosides;	Solvent extraction: acidic methanol with a concentration of 80% *v*/*v* Enzymatic extraction	-	Kromasil C18	Gradient program	HPLC-PDA 525 nm	Efficiency of different extraction methods on blueberry anthocyanins antioxidant properties	-	Antioxidant properties	[134]
58	Blueberry extracts (2021)	Dp-glycosides; Cy-glycosides; Pt-glycosides; Mv-glycosides; Pn-glycosides;	Millipore water	-	Zorbax C18 (250 × 4.6 mm, 5 µm)	Gradient program	HPLC-ESI-MS PDA 520 nm	Anthocyanins identification and their fouling mechanisms in non-thermal nanofiltration of blueberry aqueous extracts	-	-	[130]
59	Blueberries (2021)	Dp-3-glc; Cy-3-glc; Pt-3-glc; Pg-3-glc; Pn-3-glc; Mv-3-glc	1% citric acid -acidified 75% ethanol	-	Zorbax SB-C18 (250 × 4.6 mm, 5 µm)	Gradient program	HPLC-PDA 520 nm	Application of HPLC for simultaneous separation of six major anthocyanins in blueberries	-	-	[135]
60	Fresh fruits of blueberries (2022)	Dp-3,5-diglc; Dp-3-gal; Cy-3-glc; Pt-3-glc; Pg-3-glu; Mv-3-gal;	EtOH: water (7:3 *v*/*v*) mixture acidified with 1.5% HCl	Blueberries: 1107 mg/kg FW	Synergi Polar RP-18 (250 × 4.6 mm,4 µm)	Gradient program	LC-ESI-MS/MS	Simultaneous determination of anthocyanins in blueberry, strawberry and their commercial products using HPLC-MS/MS analysis	-	-	[136]
61	Highbush blueberries cultivars (2022)	Del-3-gal; Del-3-glu; Cya-3-glu; Pet-3-glu; Peo-3-glu; Mal-3-gal; Mal-3-glu;	80% MeOH containing 0.1%FA	0.3–3.2 g/kg DW	Zorbax SB-C18 (50 × 4.6 mm, 5 μm)	Gradient program	HPLC-QToF-MS	Quantification of free and bound phenolics in northern highbush blueberries	-	-	[138]
Sweet cherry (*P. avium*)											
62	Sweet cherry (2002)	Cy-3-glc; Cy-3-rut; Pn-3-glc; Pg-3-rut; Pn-3-rut;	Methanol	29–62 mg/100 g FW	Hypersil PEP 300 (250 × 4.6 mm, 5 µm)	Gradient program	HPLC-PDA PDA at 520 nm	Quantitation of anthocyanins in different cultivars of sweet cherries	-	-	[140]
63	Sweet cherry (2004)	Cy-3-glc; Cy-3-rut; Pn-3-glc; Pg-3-rut; Pn-3-rut;	60% methanol	4.9–230.3 mg/100 g FW	Novapak C18 (150 × 3.9 mm, 5 µm)	-	HPLC-PDA PDA at 520 nm	Effect of ripeness and postharvest storage on phenolic content of cherries	-	-	[141]
64	Sweet cherry (2004)	Cy-3-glc; Cy-3-rut; Pn-3-glc; Pg-3-rut; Pn-3-rut;	Methanol	0.3–116.1 mg/100 g FW	Hypersil PEP 300 (250 × 4.6 mm, 5 µm)	Gradient program	HPLC-PDA PDA at 520 nm	Changes of anthocyanins affecting skin color of sweet cherries	-	-	[142]
65	Sweet cherry (2007)	Cy-3-glc; Cy-3-rut; Pn-3-glc; Pg-3-rut; Pn-3-rut;	60% methanol	-	Novapak C18 (150 × 3.9 mm, 5 µm)	-	HPLC-PDA PDA at 520 nm	Effect of ripeness and postharvest storage on the evaluation of anthocyanins in cherries	-	-	[143]
66	Sweet cherry (2008)	Cy-3-glc; Cy-3-rut; Pg-3-rut; Pn-3-rut;	Methanol containing 1% HCl and 1% BHT	1.1–16.2 mg/100 g FW	Gemini C18 (150 × 4.6 mm, 3.0 µm)	Gradient program	HPLC-DAD-MS 520 nm	Phenolic compounds from sweet cherry	-	Antioxidant activity	[144]
67	Sweet cherry (2009)	Cy-3-glc; Cy-3-rut;	acidified methanol containing 2M HCl	26.0 mg/100 g FW	Omnispher C18 (250 × 4.6 mm, 5μm)	Gradient program	HPLC-PDA 520 nm	Phenolic composition of berries	-	Antioxidant activity	[89]
68	Sweet cherry (2014)	Cy-3-sop; Cy-3-rut; Cy-3-glc; Pg-3-rut; Pn-3-rut;	Methanol containing 1%BHA	642.5 mg/100 g FW	Luna C18 (150 × 2.0 mm, 3 µm)	Gradient program	HPLC-DAD-MS 520 nm	Metabolomic profiling of flavonoids in sweet cherry	-	-	[145]
69	Sweet cherry (2021)	Pt-3-(6-acetyl)-glc;	70% ethanol in water	Qualitative	Synergi Hydro-RP (250 × 4.6 mm, 4 µm)	Gradient program	HPLC-ESI-MS	Characterization of phenolic compounds from sweet cherry	-	Antioxidant activity	[181]
Black Chokeberry (*A. melanocarpa*)											
70	Chokeberry (2004)	Cy-3-gal; Cy-3-glc; Pg-3-gal; Cy-3-arab; Pg-3-arab; Cy-3-xyl;	Acetone/water/acetic acid (70:29.5:0.5 *v*/*v*)	-	Zorbax SB-C18 (250 × 4.6 mm, 5 µm)	Gradient program	HPLC-PDA-MS 520 nm	Characterization of anthocyanins	-	Antioxidant property	[154]
71	Chokeberry (2016)	Cy-3-gal; Cy-3-glc; Cy-3-arab; Cy-3-xyl;	80% ethanol in water	249–737 mg/100 g FW	Chromalith Performance RP18e (100 × 4.6 mm, 5 µm)	Gradient program	HPLC-DAD 520 nm	Anthocyanins content variation different Aronia cultivars	-	Antioxidant activity	[155]
72	Chokeberry (2020)	Cy-3-arab; Cy-3-gal; Cy-3-glc;	Methanol containing 6% formic acid	-	Purosphere STAR RP-18e (250 × 4.6 mm, 5 µm)	Gradient program	HPLC-DAD 520 nm	Chemometric studies of *A. melanocarpa* fruits	PCA	-	[40]
American large cranberry (*V. macrocarpon*)											
73	Cranberries extract (*V. macrocarpon*) (2004)	Cy-3-gal; Cy-3-arab; Pn-3-gal; Pn-3-arab;	Extract powder dissolved in water	-	Nova-Pak C18 (150 × 3.9 mm, 4 μm)	Gradient program	HPLC-PDA-MS 520 nm	Phytochemical constituents from total cranberry extract	-	Inhibitory effect against Hep-G2 liver cancer cells,	[188]
74	Cranberries (*V. macrocarpon*) (2011)	Cy-3-gal; Cy-3-glc; Cy-3-arab; Pn-3-gal; Pn-3-arab;	98% methanol in 2% HCl (*v*/*v*)	4.3 mg/g DW	Cosmosil 5C18-PAW (150 × 4.6 mm, 5 µm)	Gradient program	HPLC-UV/Vis 520 nm	Determination of anthocyanins in cranberry fruit	-	-	[189]
European small cranberry (*V. oxycoccus*)											
75	Cranberries (*V. oxycoccus*) (2001)	Cy-3-gal; Cy-3-glc; Pt-3-gal; Cy-3-arab; Pn-3-gal; Pn-3-arab;	Acetone/water/acetic acid (70:29.5:0.5 *v*/*v*)	Cranberries: 360 mg/100 g FW	Zorbax C18 (150 × 4.6 mm, 5 µm)	Gradient program	HPLC-MS/MS	Identification of anthocyanins and procyanidins in blueberries and cranberries	-	-	[106]
76	Cranberries (*V. oxycoccus*) (2009)	Cy-3-gal; Cy-3-glc; Cy-3-arab; Pn-3-glc; Pn-3-gal; Pn-3-arab;	Acidified 95% ethanol containing 0.1 N HCl (*v*/*v*)	40.7–207.3 mg/100 g FW	Lichrosphere C18 (125 × 4.0 mm, 5 µm)	Gradient program	HPLC-DAD-MS 520 nm	Anthocyanins in berries of European cranberry	-	Antibacterial activity	[162]
77	Fresh skin of cranberry (2016)	-	0.1 M HCl solution 90:10 (*v*/*v*)	-	Analytical C18	Gradient program	HPLC-PDA 525 nm	Mobile phase variation studies using fruit and rose extract using HPLC method		-	[123]
78	Cranberries (*V. oxycoccus*) (2021)	Cy-3-glc; Cy-3-rut; Pt-3-glc; Pn-3,5-dihex; Pg-3-glc; Pg-3-(6-malonyl)-glc;	60% ethanol at solid to liquid ration of 1:30 g/mL	-	Zorbax Eclipse XDB C18 (150 × 4.6 mm, 5 µm)	Gradient program	HPLC-PDA-MS 520 nm	Optimization of ultrasound-assisted extraction of anthocyanins from cranberries.	-	-	[190]
Black Crowberry (*E. nigrum*)											
79	Crowberry (*E. nigrum*) (2010)	Dp-glycosides; Cy-glycosides; Pt-glycosides; Mv-glycosides; Pn-glycosides;	70% aq. Acetone containing 1% formic acid	401–768.2 mg/100 g FW	XTrerra Phenyl (250 × 4.6 mm, 5 µm)	Gradient program	HPLC-DAD-MS 520 nm	Variation of anthocyanins in wild populations of crowberries	Hierarchical cluster analysis (HCA)	-	[156]
Elderberry (*S. nigra*)											
80	American and European elderberry (2007)	Cy-glycosides; Cy-acyl-glycosides;	acidified methanol (0.1% FA)	*S. canadensis*: 207.1–1005.2 mg/100 g *S. nigra*: 656.5–806.1 mg/100 g fresh weight	Synergi Hydro RP (150 × 2 mm, 4 µm)	Gradient program	HPLC-DAD-MS 520 nm	Distribution of anthocyanins in American and European elderberry varieties	-	-	[147]
81	Elderberry fruit cultivars (2009)	Cy 3-sam-5-glc; Cy 3,5-diglc; Cy 3-sam; Cy 3-glc; Cy 3-rut;	Acidified methanol containing 1% HCL and 1% 2,6-di-tert-butyl-4-methylphenol (BHT)	603–1265 mg/100 g FW	Gemini C18 (150 × 4.6 mm, 3 µm)	Gradient program	HPLC-DAD	Anthocyanin content	-	-	[216]
82	Elderberry species and hybrids (2014)	Cy-glycosides; Cy-acyl-glycosides; Pg-glycosides;	Methanol with 3% formic acid and 1% 2,6-di-tert-butyl-4-methylphenol (BHT)	560 mg/100 g FW	Gemini C18 (150 × 4.6 mm, 3 µm)	Gradient program	HPLC-DAD-ESI-MS	Anthocyanin analysis in different accessions of elderberry fruits	-	-	[163]
83	Elderberry fruit, other species (2022)	Cy-3-sam-5-glc; Cy-3-sam; Cy-3-glc	acidified methanol (1% formic acid)	0.08–5.3 mg/g DW	HSS C18 (100 × 2.1 mm, 1.8 µm)	Gradient program	UHPLC-PDA-MSUHPLC-QToF-MS	Authentication, characterization of anthocyanins and other polyphenolic compounds	-	-	[182]
Goji berry/Wolf berry (*L. ruthenicum*)											
84	Black Goji berry (2003)	-	Various solvents such as petroleum ether, ethyl acetate, methanol and n-butanol inn individual	Tr-3.8 mg/g DEW	Discovery C18 (250 × 4.6 mm, 5 µm)	Gradient program	HPLC-PDA	Antioxidant activity of Lycium extracts	-	Antioxidant activity	[148]
85	Black Goji berry (2011)	Pt-glycosides; Pt-acyl-glycosides;	Methanol containing 2% formic acid	465–525 mg/100 g FW	ODS 80Ts QA (150 × 4.6 mm, 5 µm)	Gradient program	HPLC-DAD and HPLC-ESI-MS 520 nm	Anthocyanin composition of goji berries from Qinghai-Tibet Plateau	-	Antioxidant activity	[149]
86	Black Goji berry (2015)	Pt-acyl-glycosides; Dp-acyl-glycosides; Mv-acyl-glycosides;	70% ethanol pH 2.5 adjusted with HCl	Qualitative	Xterra MS C18 (150 × 4.6 mm, 5 µm)	Gradient program	HPLC-DAD 520 nm LC-QToF-MS	Characterization of anthocyanins from black goji berry	-	-	[212]
87	Black Goji berry (2016)	Dp-glycosides; Cy-glycosides; Pt-glycosides; Mv-glycosides; Pg-glycosides; Acyl-glycosides	2% formic acid in aq. Solution	9.3–37.8 mg/g DW	ODS 80Ts QA (150 × 4.6 mm, 5 µm)	Gradient program	HPLC-DAD-MS 520 nm	Constituent analysis and quality control of anthocyanins in dried goji berries	-	-	[198]
88	Black Goji berry (2018)	Dp-glycosides; Pt-glycosides; Mv-glycosides; Pg-glycosides; Acyl-glycosides	Deep eutectic solvent (DES) extraction: various molar ratios of choline chloride: 1,2-propanediol (mol/mol)	3.43–3.83 mg/g FW	Zorbax SB-C18 (100 × 4.5 mm, 3.5 µm)	Gradient program	2D HPLC-DAD 520 nm LC-QToF-MS	DES based extraction for the determination of anthocyanins in goji berry using 2D LC-DAD-ESI-MS/MS	-	-	[184]
89	Black Goji berry (2022)	Dp-glycosides; Pt-glycosides; Mv-glycosides; Acyl-glycosides	2% formic acid in methanol	1485.2–5826.0 mg/100 g DW	Zorbax SB-C18 (250 × 4.5 mm, 5 µm)	Gradient program	HPLC-PDA 525 nm LC-QToF-MS	Anthocyanin fingerprint of black wolfberry fruit	PCA	-	[183]
Red grapes Peel (*V. vinifera*)											
90	Red grapes skin (2001)	Dp-glycosides; Pt-glycosides; Mv-glycosides; Pn-glycosides; acyl-glycosides;	-	Relative quantification (%)	Waters Novapak (150 × 3.9 mm, 5 µm)	Gradient program	HPLC-PDA 520 nm	HPLC analysis of anthocyanins in different red grape cultivars	-	-	[213]
91	Red grapes skin (2002)	Dp-glycosides; Cy-glycosides; Pt-glycosides; Mv-glycosides; Pn-glycosides; acyl-glycosides;	-	Relative quantification (%)	Waters Novapak (150 × 3.9 mm, 5 µm)	Gradient program	HPLC-PDA 520 nm	Anthocyanin patterns of red grape cultivars	-	-	[211]
92	Red grapes skin (2004)	Dp-glycosides; Cy-glycosides; Pt-glycosides; Mv-glycosides; Pn-glycosides; acyl-glycosides;	Methanol	-	Supersphere 100 RP 18 (250 × 4.6 mm, 5 µm)	Gradient program	HPLC-DAD-MS 520 nm	Anthocyanins identification in red grapes skin by LC-MS and NMR	HCA and RDA	-	[197]
93	Red wine grapes genotypes (2004)	Dp-glycosides; Cy-glycosides; Pt-glycosides; Mv-glycosides; Pn-glycosides; acyl-glycosides;	methanol/water/FA (60:37:3 *v*/*v*)	Red wine grapes: 0.38–7.9 g/kg FW	Symmetry C18 (250 × 4.6 mm, 5 μm)	Gradient program	HPLC-PDA-MS PDA at 520 nm	Flavonoid glycosides estimation in various genotypes of red wine grapes, blackberry and blueberries	-	Antioxidant properties	[83]
94	Red grapes skin/peel (2005)	Dp-glycosides; Cy-glycosides; Pt-glycosides; Mv-glycosides; Pn-glycosides; acyl-glycosides;	20% formic acid in methanol	-	Kromasil 100 C18 (250 × 4.0 mm, 4 µm)	Gradient program	HPLC-PDA 546 nm	Varietal difference among anthocyanins in red grape cultivars	PCA	-	[214]
95	Red grapes skin (2008)	Cy-3,5-diglc; Cy-3-glc; Mv-3-glc; Mv-3,5-diglc; Dp-3-glc; Pn-3-glc; Pg-3-diglc; Pt; Pg;	Acidified methanol (1% HCl *v*/*v*)	3.4–7.1 mg/g FW	Inertsil ODS-3V (150 × 4.6 mm, 5 µm)	Gradient program	RP-HPLC-ED	Anthocyanin monitoring in red grape skin extracts	-	-	[185]
96	Red grapes skin (2008)	Dp-3-glc; Cy-3-glc; Pn-3-glc; Mv-3-glc; Pt-3-glc;	Pressurized fluid extraction: methanol	564.7 mg/g DW	Synergi C12 Max-RP (250 × 4.6 mm, 4 µm)	Gradient program	HPLC-UV/Vis 520 nm	Determination of anthocyanins in red grape skin	-	-	[79]
97	Red grapes skin (2008)	Dp-glycosides; Cy-glycosides; Pt-glycosides; Mv-glycosides; Pn-glycosides; acyl-glycosides;	Acidic ethanol (0.1% HCl)	1.7–4.1 mg/g FW	Eurosphere C18 (250 × 4.6 mm, 5 µm)	Gradient program	HPLC-PDA-MS 530 nm	Extraction methods for separation of anthocyanins from red grape skin	-	-	[186]
98	Red grapes skin (2012)	Dp-glycosides; Cy-glycosides; Pt-glycosides; Mv-glycosides; Pn-glycosides; acyl-glycosides;	Methanol/water/formic acid (50:48.5:0.5 *v*/*v*)	2.4–25.3 mg/100 g FW	Zorbax XDB-C18 (250 × 4.6 mm,5 µm)	Gradient program	HPLC-DAD	Phenolic composition of red grape varieties from Spanish region	-	-	[157]
99	Red grapes skin (2013)	Dp-glycosides; Cy-glycosides; Pt-glycosides; Mv-glycosides; Pn-glycosides; acyl-glycosides;	Buffer composition (1 L): 200 mL deionized water, 5 g tartaric acid, 120 mL of 95 % ethanol, 2 g sodium metabisulfite, 22 mL of a 1 N sodium hydroxide solution, and then, deionized water up to the 1 L volume.	6.9–45.0 mg/100 g FW	Hypersil ODS (200 × 2.1 mm, 5 µm)	Gradient program	HPLC-DAD 520 nm	Changes in phenolic content during ripening	PCA	-	[165]
100	Red grapes skin (2014)	Dp-glycosides; Cy-glycosides; Pt-glycosides; Mv-glycosides; Pn-glycosides; acyl-glycosides;	Methanol containing 0.5% 12N HCl	-	Phenomenex Jupiter C18 (250 × 2.1 mm, 4 µm)	Gradient program	HPLC-DAD 520 nm MALDI-Tof-MS LC-QToF-MS	Profiling of anthocyanins in grape varieties	-	-	[215]
101	Red grapes skin (2015)	Dp-3-glc; Cy-3-glc; Pt-3-glc; Pg-3-glc; Mv-3-glc; Pn-3-acetyl-glc; Cy; Pn-3-coumaroyl-glc; Pn; Mv;	Distilled water	7–24.3 mg/g DEW	Zorbax Eclipse plus C18 (150 × 4.6 mm, 3.5 µm)	Gradient program	HPLC-DAD 520 nm	Anthocyanins and total phenolics profile of red grape varieties.	-	Antioxidant activity	[187]
Black Mulberry (*M. nigra*)											
102	Mulberry (2001)	Cy-3-sop; Cy-3-glc; Cy-3-rut; Pg-3-glc; Pg-3-rut;	Aliquot of 5 mL of extract from fresh berries was taken and centrifuged	-	Restek Pinnacle ODS (250 × 4.6 mm, 5 μm)	Gradient program	HPLC-ESI-MS	Identification of anthocyanins in different berries	-	-	[82]
103	Mulberry (2010)	Cy-3-rut; Cy-3-glc; Pg-3-glc; Pg-3-rut; Cy; Pg;	-	Qualitative	Waters RP C18 (250 × 4.6 mm, 5 µm)	Gradient program	HPLC-PDA-MS 520 nm	Analysis and characterization of anthocyanins from mulberry	-	-	[209]
104	Mulberry (2011)	Cy-3-(2-glucosyl)-rut; Dp-3-rut-5-glc; Cy-3,5-diglc; Cy-3-glc; Cy-3-rut; Pg-3-glc; Pg-3-rut; Dp-3-rut;	Acidified methanol concentration varied from 10–70% containing 1%TFA under microwave assisted extraction conditions	-	Zorbax SB-C18 (50 × 2.1 mm, 1.8 µm)	Gradient program	HPLC-ESI-MS	Optimization of microwave extraction for anthocyanins from mulberry and analysis using HPLC-ESI-MS	-	-	[164]
105	Mulberry (2017)	Dp-3-rut; Cy-3-glc; Cy-3-rut; Cy-3-rut-5-glc;	0.1%HCl in ethanol	0.49 mg/mL	Zorbax Eclipse plus C18 (100 × 4.5 mm, 3.5μm)	Ethanol and α-hydroxy acid aqueous solution	HPLC-DAD 520 nm HPLC-DAD-MS	Green chromatography for anthocyanins determination in berries	-	-	[98]
106	Mulberry (2020)	Dp-pent; Pt-pent; Pn-hex; Cy-pent-hex; Cy-rha-hex; Cy-sambu-glc; Dp-dirha-hex;	1% formic acid in methanol	-	Varian pursuit C18 (150 × 2.0 mm, 3 µm	Gradient program	HPLC-DAD-MS 520 nm	Identification of phenolic compounds in edible wild fruits using HPLC-DAD-ESI-HRMS	-	-	[210]
107	Mulberry (2020)	Cy-3-glc; Cy-3-rut; Pg-3-glc; Pg-3-rut;	acetone containing 1% HCl	125.3 mg/100 g DEW	Zorbax Eclipse plus C18 (150 × 4.6 mm, 5 μm)	Gradient program	HPLC-DAD 520 nm	Comparative HPLC analysis of Morus species	-	-	[146]
Peach (yellow peel) (*P. persica*)											
108	Peaches (2001)	Cy-3-glc; Cy-3-rut;	water: methanol (20:80) containing sodium formate	6.9–33.6 mg/100 g FW	Nucleosil C18 (150 × 4.6 mm, 5 µm)	Gradient program	HPLC-DAD-MS 510 nm	Phenolic compounds from peel of peaches and plums	-	-	[199]
109	Peaches (2021)	Cy-3-glc; Cy-3-rut;	methanol/water/0.1 M HCl (6:3:1 *v*/*v*)	0.5–2.8 mg/100 g FW	Inertsil ODS-3 (250 × 4.6 mm, 5 µm)	Gradient program	HPLC-DAD 520 nm	Polyphenols content in peaches and plums	PCA	Antioxidant activity	[205]
Plum (red and black peel) (*P. domestica*)											
110	Plums peel (2001)	Cy-3-glc; Cy-3-rut; Cy-3-acetyl-glc; Cy-3-gal;	water: methanol (20:80) containing sodium formate	12.9–161.4 mg/100 g FW	Nucleosil C18 (150 × 4.6 mm, 5 µm)	Gradient program	HPLC-DAD-MS 510 nm	Phenolic compounds from peel of peaches and plums	-	-	[199]
111	Plums (2006)	Dp-3-sambu; Cy-3-sambu; Pn-3-sambu; Pt-3-sambu;	80% methanol containing 0.1% HCl (*v*/*v*)	Davidson’s plum: 1.27 µM/g FW Lllawarra plum: 19.4 µM/g FW Burdekin plum: 6.0 µM/g FW	Luna C18 (250 × 4.6 mm, 5 µm)	Gradient program	HPLC-DAD-MS 520 nm	Identification and quantification of anthocyanins in Australian native fruits	-	Antioxidant activity	[203]
112	Plums (2009)	Cy-3-xyl; Cy-3-glc; Cy-3-rut; Pn-3-rut; Pn-3-glc;	methanol containing 1% HCl and 1% BHT	5.3–18.7 mg/100 g FW	Phenomenex gemini C18 (150 × 4.6 mm, 3 µm)	Gradient program	HPLC-PDA 530 nm	Anthocyanins and fruit color in plums	-	-	[200]
113	Plums (2021)	Cy-3-glc; Cy-3-rut; Pn-3-rut;	methanol/water/0.1 M HCl (6:3:1 *v*/*v*)	1.2–16.4 mg/100 g FW	Inertsil ODS-3 (250 × 4.6 mm, 5 µm)	Gradient program	HPLC-DAD 520 nm	Polyphenols content in peaches and plums	PCA	Antioxidant activity	[205]
Pomegranate (*P. granatum*)											
114	Pomegranate peel (2011)	Dp-3,5-diglc; Cy-3,5-diglc; Pg-3,5-diglc; Dp-3-glc; Cy-pent-hex; Cy-3-glc; Cy-3-rut; Pg-3-glc; Cy-pent;	aqueous methanol (80% *v*/*v*; 0.15 M HCl)	Peel: 44.7 mg/100 g DW	Synergi Hydro-RP (150 × 3.0 mm, 4 µm)	Gradient program	HPLC-DAD-MS 520 nm	Identification and quantification of phenolic compounds from pomegranate	-	Antioxidant activity	[175]
115	Pomegranate peel (2013)	Dp-3-glc; Cy-3-glc; Pg-3-glc; Pn-hex; Cy-pent;	Methanol (0.1% HCl *v*/*v*)	5.3–102.9 mg/100 g FW	Zorbax SB-C18 (150 × 4.6 mm, 5 µm)	Gradient program	HPLC-DAD-MS 520 nm	Characterization and evaluation of major anthocyanins in pomegranate peel from different cultivars	-	-	[201]
116	Pomegranate peel (2015)	Dp-3,5-diglc; Dp-3-glc; Cy-3,5-diglc; Cy-3-glc; Pg-3,5-diglc; Pg-3-glc;	Methanol (0.1% HCl)	45.2–344.1 mg/100 g FW	Zorbax SB-C18 (150 × 4.6 mm, 5 µm)	Gradient program	HPLC-DAD 520 nm	Composition and content of anthocyanins in pomegranate cultivars	-	-	[207]
117	Pomegranate peel (2015)	Dp-3,5-diglc; Dp-3-glc; Cy-3,5-diglc; Cy-3-glc; Pg-3,5-diglc; Pg-3-glc;	Methanol (0.1% HCl)	3.7 mg/100 g FW	Zorbax SB-C18 (150 × 4.6 mm, 5 µm)	Gradient program	HPLC-DAD 520 nm	Patterns of pigment changes in pomegranate peel during fruit ripening	-	-	[208]
118	Pomegranate peel (2016)	-	Peel extraction (Soxhlet): Ethyl acetate	-	Lichrosorb RP 18 (250 × 4.6 mm, 5 µm)	Gradient program	HPLC-DAD	Evaluation of extraction methods from pomegranate whole fruit or peel	-	Antioxidant, antiproliferative activity and cytotoxicity assay	[176]
119	Pomegranate peel (2017)	Cy-3,5-diglc; Cy-3-glc; Pg-3-glc;	different concentration of ethanol.	-	BDS Hypersil (250 × 4.6 mm, 5 µm)	Gradient program	HPLC-DAD-MS 520 nm	High-Pressure-Assisted Extraction of Bioactive Compounds from Pomegranate Peel	Response surface methodology (RSM)	Antioxidant activity	[177]
120	Pomegranate peel (2017)	Dp-3,5-diglc; Dp-3-glc; Cy-3,5-diglc; Cy-3-glc; Pg-3,5-diglc; Pg-3-glc;	water/methanol (80:20 *v*/*v*) containing 0.1% HCl.	-	Ascentis express C18 (250 × 4.6 mm, 5 µm)	Gradient program	HPLC-PDA-MS 520 and 540 nm	Metabolite fingerprinting of pomegranate polyphenols using HPLC-DAD-MS	PCA	-	[178]
121	Pomegranate peel (2017)	Dp-3,5-diglc; Cy-3,5-diglc; Cy-3-glc; Pg-3,5-diglc; Pg-3-glc;	Soxhlet condition using ethanol as solvent	7.9–10.3 mg/100 g DW	μBondapack (200 × 4.6 mm, 10 µm)	Gradient program	HPLC-PDA 520 nm	Phenolic compounds from pomegranate peel	-	Antioxidant activity	[179]
122	Pomegranate peel (2020)	Dp-3,5-diglc; Cy-3,5-diglc; Dp-3-glc; Cy-3-glc; Pg-3-glc;	Millipore water	Peel: nd–68 mg/100 g FW	Kinetex EC-C18 (30 × 3 mm, 2.6 µm)	Gradient program	HPLC-PDA-MS 520 nm	Characterization of peel from pomegranate	-	Antioxidant activity, α-amylase inhibitory activity and tyrosine inhibitory activity	[180]
Raspberry (red) (*R. idaeus*)											
123	Red raspberries (2002)	Cy-3,5-diglc; Cy-3-glc	0.1%HCl acidified methanol	-	Phenomenex Phenyl-hexyl (250 × 4.6 mm, 5 µm)	Gradient program	HPLC-MS	Phenolic content of berries (not LC based)	-	Antioxidant activity	[15]
124	Red raspberries (2002)	Cy-glycosides; Pg-glycosides; acyl-glycosides;	0.1% HCl in methanol	-	RP-MAX (250 × 4.6 mm, 4 µm)	Gradient program	HPLC-MS	Chemical composition of red raspberries	-	Antioxidant and Vasorelaxation activity	[169]
125	Red raspberries (2002)	Cy-3-sop; Cy-3-(2-glucosyl)-rut; Cy-3-glc; Pg-3-sop; Cy-3-rut; Pg-3-(2-glucosyl)-rut; Pg-3-glc; Pg-3-rut;	0.1% HCl in methanol	-	NovaPac C18 (250 × 4.6 mm, 5 µm)	Gradient program	HPLC-PDA-MS 520 nm	Characterization of anthocyanins in red raspberries	-	-	[170]
126	Red raspberries (2004)	Cy-3-hex; Cy-3-sop; Cy-3-(2-glucosyl)-rut; Cy-3-glc; Cy-3-rut; Pg-3-glc; Pg-3-rut; Cy;	ethyl acetate followed by acidified methanol	53.6–88.8 mg/100 g FW	Lichrocart Purospher RP-18e (125 × 3 mm, 5 µm)	Gradient program	HPLC-DAD-MS 520 nm	Identification and quantification of phenolic compounds	-	-	[152]
127	Red raspberries (2007)	Cy-3-sop; Cy-3-(2-glucosyl)-rut; Cy-3-sambu; Cy-3-glc; Cy-3-xylosyl-rut; Cy-3-rut; Pg-3-rut;	1.5 M HCl/95% ethanol (15:85 *v*/*v*)	-	Kromasil C18 (250 × 4.6 mm, 5 µm)	Gradient program	HPLC-MS	Optimization of ultrasound assisted extraction for anthocyanin extraction and their identification	-	-	[158]
128	Raspberry (2007)	Dp-glycosides; Cy-glycosides; Pt-glycosides; Mv-glycosides; Pn-glycosides;	1 N HCl acidified methanol (85:15 *v*/*v*)—pH to 1.0	365.2 mg/100 g DW	Luna C18 (150 × 3 mm, 3 µm)	Gradient program	HPLC-PDA 520 nm and UPLC-ESI-MS	Anthocyanin content determination in berries	-	-	[153]
129	Red Raspberry (2009)	Cy-3-sop; Cy-3-glc;	Acidified methanol containing 2 M HCl	77.2 mg/100 g FW	Omnispher C18 (250 × 4.6 mm, 5 μm)	Gradient program	HPLC-PDA 520 nm	Phenolic composition of berries	-	Antioxidant activity	[89]
130	Raspberries (2010)	Cy-glycosides; Pg-glycosides; acyl-glycosides;	methanol with 0.1% HCl	Qualitative	Synergi RP-Max (250 × 2 mm, 4 µm)	Gradient program	HPLC-DAD-MS/MS	Anthocyanins analysis in berries	-	-	[117]
131	Raspberries (2010)	Cy-3-sop; Cy-3-(2-glc)-rut; Cy-3-glc; Cy-3-rut; Cy-3,5-diglc;	0.1% formic acid in methanol	76.2–277.0 mg/100 g DW	Nova-Pak C18 (150 × 3.9 mm, 4 µm)	Gradient program	HPLC-DAD 520 nm	HPLC analysis of polyphenols from Red raspberries	-	-	[159]
132	Raspberry (2016)	Cy-3-glc; Pt-3-glc; Cy-3-sop; Cy-3-glucosyl-rut; Cy-3-rut;	-	133.9 mg/100 g FW	Lichrocart RP-18 (250 × 4.6 mm, 5 μm)	Gradient program	HPLC-DAD 520 nm	Phenolic composition and antioxidant activity of different berries	-	Antioxidant activity	[97]
133	Red raspberry (2018)	Cy-3-sop; Cy-3-(2-glucosyl)-rut; Cy-3-glc; Cy-3-rut;	1 N HCl in 75% methanol	188.0 mg/100 g DW	Zorbax SB-C18 (250 × 4.6 mm, 5 µm)	Gradient program	HPLC-PDA-MS 520 nm	Comparative study of red berry pomaces	stepwise linear discriminant analysis (SLDA)	Antioxidant activity	[102]
134	Red raspberry (2020)	Cy-3-sop; Cy-3-glc;	ethanol/water mixtures acidified with citric acid until pH 3	613–1000 mg/100 g DEW	Phenomenex Aqua RP-C18 (150 × 4.6 mm, 5 µm)	Gradient program	HPLC-DAD-MS 520 nm	Anthocyanins extraction from red raspberry and their activities	Response surface graphs	Antioxidant activity and Antibacterial effect	[161]
135	Red raspberry (2020)	Cy-3-sop; Cy-3-glc; Cy-3-glucosyl-rut; Cy-3-rut; Pg-3-glc;	95% acidified ethanol (pH 3.0)	1.7–102.5 mg/100 g FW	Thermo Hypersil Gold C18 (100 × 2.1 mm, 3 µm)	Gradient program	HPLC-DAD-MS 520 nm	Physicochemical characteristic evaluation for Red raspberries	PCA(Principle component analysis)	-	[160]
136	Red raspberry (2021)	Cy-3-glc; Cy-3-rut; Cy-3-xylosyl-rut;	80:20 ethanol/water (*v*/*v*)	336–554 mg/100 g DEW	-	-	HPLC-MS	Antibacterial effects of berries against H. pylori infection	-	Antibacterial activity	[132]
Raspberry (black) (*R. occidentalis*)											
137	Black raspberries (2002)	Cy-3-(6-p-coumaroyl)-sambu; Cy-3-(6-p-coumaroyl)-glc;	0.1%HCl acidified methanol	-	Phenomenex Phenyl-hexyl (250 × 4.6 mm, 5 µm)	Gradient program	HPLC-MS	Phenolic content of berries (not LC based)	-	Antioxidant activity	[15]
138	Black raspberries (2005)	Cy-3-glc; Cy-3-sambu; Cy-3-(2-xylosyl)-rut; Cy-3-rut	Methanol	-	Symmetry C18 (75 × 4.6 mm, 3.5 µm)	Gradient program	LC-MS/MS	Anthocyanins determination in black raspberries	-	-	[206]
139	Black raspberry (2021)	Cy-3-glc; Cy-3-rut; Cy-3-xylosyl-rut;	80:20 ethanol/water mixture	2885–11109 mg/100 g DEW	-	-	HPLC-MS	Antibacterial effects of berries against H. pylori infection	-	Antibacterial activity	[132]
Strawberry (*Fragaria × ananassa*)											
140	Strawberry (2002)	Cy-glycosides; Pg-glycosides; acyl-glycosides;	0.1%HCl in methanol	-	Phenomenex Aqua C18 (150 × 4.6 mm, 5 µm)	Gradient program	HPLC-PDA-MS 520 nm	Identification of anthocyanins in strawberry	-	-	[77]
141	Strawberry (2004)	Cy-3-glc; Cy-glc; Pg-3-glc; Pg-3-rut; Cy; Pg-3-malonyl-glc; Pg-3-succinyl-glc;	acidified methanol	31.4–36.5 mg/100 g FW	Lichrocart Purospher RP-18e (125 × 3 mm, 5 µm)	Gradient program	HPLC-DAD-MS 520 nm	Identification and quantification of phenolic compounds	-	-	[152]
142	Strawberry (2004)	Cy-3-glc; Pg-3-glc; Pg-3-ara; Pg	acidified methanol	25.3–39.8 mg/100 g FW	Lichrocart Purospher RP-18 (125 × 3 mm, 5 µm)	Gradient program	HPLC-PDA 520 nm	Comparison of different strawberry cultivars in Poland	-	Antioxidant activity	[151]
143	Strawberries (2007)	Dp-3-glc; Dp-3-rut; Dp-3-gal; Cy-3-glc; Mv-3-glc; Pn-3-glc; Pn-3-gal; Mv-3-gal; Mv-3-ara;	1 N HCl acidified methanol (85:15 *v*/*v*)—pH to 1.0	97.5 mg/100 g DW	Luna C18 (150 × 3 mm, 3 µm)	Gradient program	HPLC-PDA 520 nm and UPLC-ESI-MS	Anthocyanin content determination in berries	-	-	[153]
144	Strawberry (2008)	Cy-glycosides; Pg-glycosides; acyl-glycosides;	1% acetic acid in acetone/1% acetic acid in methanol (1:1 *v*/*v*)	Qualitative	Eclipse XDB C18 (150 × 4.6 mm, 5 µm)	Gradient program	HPLC-DAD-MS 520 nm	Identification of phenolic compounds in cultivated strawberries from Macedonia	-	-	[174]
145	Strawberry (2011)	Cy-3-glc; Cy-3-rut; Pg-3-glc; Pg-3-rut; Pg-3-malonyl-glc; Pg-3-acetyl-glc	acetone/water/acetic acid-70:29.5:0.5 *v*/*v*	6.6–45.3 mg/100 g FW	Ultrasphere ODS (250 × 4.6 mm, 5 µm)	Gradient program	HPLC-PDA 520 nm	Characterization of phenolic compounds in strawberry fruits	-	Antioxidant activity	[173]
146	Strawberry (2012)	Cy-3-glc; Pg-3-glc; Pg-3-rut; Pg-3-malonyl-glc; Pg-derivative;	acidified methanol (0.1% HCl *v*/*v*)	269.2–559.4 mg/100 g DW	Sunfire C18 (250 × 4.6 mm, 5 µm)	Gradient program	HPLC-DAD-MS 520 nm	Evaluation of freezing and thawing methods on strawberry anthocyanin and color stability	-	Antioxidant activity	[172]
147	Strawberry (2013)	Cy-3-glc; Pg-3-glc; Pg-3-rut; Pg-3-malonyl-glc;	acetone/water/acetic acid-70:29.5:0.5 *v*/*v*	14.6–34.3 mg/100 g FW	Lichrocart C18 (250 × 4 mm, 5 μm)	Gradient program	HPLC-DAD-MS 520 nm	Bioactive compounds from strawberries	-	Antioxidant activity	[166]
148	Strawberry (2013)	Cy-3-glc; Pg-3-glc; Pg-3-rut; Pg-3-malonyl-glc; Pg-derivative;	acidified methanol (0.1% HCl *v*/*v*)	4.7–31.7 mg/100 g FW	Sunfire C18 (250 × 4.6 mm, 5 µm)	Gradient program	HPLC-DAD-MS 520 nm	Influence of PPO inhibitors on strawberry anthocyanin and color stability	-	Antioxidant activity	[167]
149	Strawberry (2015)	Cy-glycosides; Pg-glycosides; acyl-glycosides;	methanol containing 1%HCl *v*/*v*	212.2–758.6 mg/100 g DW	Zorbax SB-C18 (250 × 4.6 mm, 5 µm)	Gradient program	HPLC-DAD-MS 520 nm	Assessment of differences in phenolic compositions of strawberry cultivars	Forward Linear Discriminant Analysis (LDA)	-	[171]
150	Strawberry (2016)	Cy-3-rut; Pg-3-glc; pg-3-rut; Pn-3-rut;	-	407.8 mg/100 g DW	Lichrocart RP-18 (250 × 4.6 mm, 5μm)	Gradient program	HPLC-DAD 520 nm	Phenolic composition and antioxidant activity of different berries	-	Antioxidant activity	[97]
151	Strawberry (2020)	Cy-3-glc; Pg-3-glc;	acidified methanol (0.1% HCl *v*/*v*)	11–49.3 mg/100 g FW	Inertsil ODS-3 (250 × 4.6 m, 5 µm)	Gradient program	HPLC-DAD 520 nm	Anthocyanin content and physicochemical properties of strawberry cultivars in anthocyanin content rich	-	-	[168]
152	Fresh fruits of strawberries (2022)	Strawberries: Cya-3-glu; Pg-3-rut; Pg-3-glu;	EtOH: water (7:3 *v*/*v*) mixture acidified with 1.5% HCl	Strawberries: 20 mg/100 g FW	Synergi Polar RP-18 (250 × 4.6 mm, 4 μm)	Gradient program	LC-ESI-MS/MS	Simultaneous determination of anthocyanins in blueberry, strawberry and their commercial products using HPLC-MS/MS analysis	-	-	[136]

The highest levels of delphinidin derivatives are found in blackcurrant (392 mg/100 g), while the highest levels of peonidin derivatives are found in American cranberry (36 mg/100 g). Pelargonidin is principally found in strawberries and is mainly present as Pg-3-glc and Pg-3-(6-suc-glc) (total 58 mg/100 g). The anthocyanins in strawberries were characterized by LC using DAD and ESI-MS, and 15 compounds were identified [77]. A more complex anthocyanin profile is found in blueberries. Twenty-five different anthocyanins were identified in the highbush and the lowbush varieties of blueberries [217]. Polyphenol content is generally higher in lowbush than in highbush blueberries [217,218]. The major anthocyanins in blueberries are conjugated forms of malvidin (respectively 76 and 48 mg/100 g in lowbush and highbush blueberry), delphinidin (46 mg/100 g in both lowbush and highbush blueberry), petunidin (29 mg/100 g in both lowbush and highbush blueberry) and cyanidin (24 and 9.9 mg/100 g in lowbush and highbush blueberry, respectively). Some acetylated anthocyanins have been detected in lowbush and highbush cultivars, but are absent in others [106,217,218,219]. Table 5 shows the anthocyanin content of commonly consumed berries and fruits using various analytical methods. Many factors, including the geographic location, environmental factors, genotypes, growing and collection season, cultivar, ripening stage, processing, storage conditions, extraction solvent, time, and temperature influence the total content and yields of anthocyanins.

Analyzing the anthocyanins in fruit or berries before use in the food processing (juice, wine, jam, etc.) industry or in dietary supplements will allow better decision-making and improve the quality. Without having analytical data on composition, the end product will leave a detrimental impact on quality, and sometimes these analytical results will help to make correct harvesting and selection decisions. Figure 3A shows the contributions according to analytical methods and Figure 3B shows the contributions according to the fruit/berries determined using five different analytical techniques. Based on these pie charts, the LC is widely used and an advisable choice. Among the most studied berries/fruit were blueberry (17%), raspberry (11%) and red grape (12%).

## 5. Adulteration Issues of Processed Food or Dietary Supplements

Anthocyanins, found in fruits and berries, such as pomegranates, red grapes and wine, blueberry, cranberry, bilberry, elderberry, raspberry, and mulberry, have led to the possibility of deliberate adulteration with similar but less expensive fruit/berries (economically motivated adulteration) or synthetic ingredients, dilution with water, the addition of sugar solution, citric or tartaric acids, mislabeling the botanical or geographical origin, or addition of colorants which may not provide the desired beneficial health effects. These are widely marketed as foods, juices, and food or dietary supplements with the quality content of anthocyanins depending on storage, purification, and processing.

The increase in demand exceeding the supply makes these products vulnerable to adulteration. Overall, adulteration impacts the product’s quality, deceives consumers, and sometimes may involve health risks as the undeclared chemical constituents might interfere with medication and cause allergic reactions or other adverse effects. Different chemical methods have been reported for adulterating commercial herbal products sold in different countries.

The most important sources of beneficial anthocyanins are bilberry, elderberry, chokeberry, tart cherries, etc. Because of the growing interest and demand for berry-based extracts, supplements, and products, there has been a dire need for efficient quality control measures to ensure the authenticity of the anthocyanin sources and contents in these products and to verify label claims for compliance. The most common method used in the nutritional food supplement industry is quantifying the total anthocyanins in these raw materials spectrophotometrically at 520 nm at a controlled pH [220,221]. Spectrophotometric determination of total anthocyanins without hydrolysis is commonly adapted and practiced because of its relatively milder conditions, rapidness, and cost-effective nature [221]. This method works very well where an estimation is needed rather than an accurate quantification. Therefore, such a method is an excellent tool for rapid screening of total anthocyanin contents in fruits and vegetables. Spectral similarities in the majority of anthocyanins makes this determination relatively feasible, and the anthocyanin reported as “external standard equivalents” in most cases is “Cy-3-glc”.

The full-text articles were assessed and screened based on the inclusion and exclusion criteria (1) herbal products or commercial extracts, (2) the analyzed products (juice/concentrate, jam, supplements, wine) have to be either purchased or bought but excluded the samples obtained cost-free or as gifts, (3) “authentic” or “adulterated” had to be present for the analyzed studies, (4) analysis using a chemical method, (5) papers published in other than English were not referenced in this review and finally (6) papers using the ‘prepared adulterants’ were not discussed. A total of 27 relevant articles related to adulteration or authentication that fit the inclusion and exclusion criteria were considered and summarized accordingly. Most of the products were reported to be authentic (60–75%), but more than a quarter proved to be adulterated (25–30%) based on the botanical identity of their content compared to the ingredients stated on the labels. Anthocyanin profiles have been successfully used in quality control since specific berries have their typical anthocyanin profiles. The use of multiple techniques and markers in the authentication is necessary to achieve the highest possible certainty level and to combat fraudulent practices.

Most of the berry-containing products (bilberry, elderberry, açai, strawberry, pomegranate, raspberry, blueberry, and cranberry) are widely marketed as food, juice, jam, concentrate, and as dietary supplements. Apart from their anthocyanin profiles, the anthocyanin composition may also be altered during processing and storage. For example, marked changes and degradation of anthocyanins were observed during the processing of blueberries and strawberries into juices and concentrates [222,223]. The extent of degradation is larger in concentrates than in juices. In particular, enzymatic processing may alter anthocyanin profiles [224], and the quality content of anthocyanins from juices, jams, or other preparations depends on storage, purification, and processing.

Anthocyanins may be used as qualitative and quantitative markers in the authenticity of fruit or berry-based products. However, using anthocyanin profiles to determine authenticity will be complicated if the product contains multiple fruits with unknown ratios. The chemical profiles of anthocyanins derived from specific berries or fruits have been employed as references to identify and authenticate a blend of berries or anthocyanin-rich fruits using various chromatographic techniques (Table 6).

In the present review, the detection of adulteration was established using different techniques including FTIR spectroscopy (n = 2), thin-layer chromatography (TLC, n = 1), NMR (n = 2), high-performance liquid chromatography with UV and (or) MS (HPLC, n =17), or with chemometric analysis (n = 6). The advantages of these methods depend on the presence of specific markers that might be present in either adulterant or pure samples. HPLC is the most useful and commonly used technique for analyzing anthocyanins. NMR, IR, TLC, and CE applications to assess authenticity or detect adulteration of fruits and fruit-containing products are limited to liquid samples such as juices, extracts and supplements.

Penman et al. [225] found a synthetic dark red-blue dye, amaranth, in commercial bilberry extracts with the color being similar to the color of bilberry extracts. The adulterant was identified using HPLC-MS and NMR. Cassinese et al. [226] studied bilberry extract products using LC-MS and reported that only 15% of 40 finished bilberry products met the stated label claims for anthocyanin (ACN) content, and 10% were devoid of ACNs. 25% of the extracts differed from that of the typical bilberry, and some of these exhibited a higher content of anthocyanidins, which is an index of ACN degradation due to incorrect processing or storage conditions. Thus, ACN identification is critical in adulteration studies and in evaluating the quality of crude and processed food. This study also showed that 8 of 15 (USA), 2 of 2 (Italy), 1of 1 (Malaysia), 0 of 20 (Japan) were adulterated with berries different from *V. myrtillus* and found differences in marked compositions. A similar compliance study was conducted by Artaria et al. [227], using the same HPLC-UV method. In this study it was found that 5 of 10 commercial supplements containing bilberry extracts were noncompliant under the labeling-compliance terms for anthocyanin content. Gardana et al. [228] analyzed quality of bilberry fruit as well 14 extracts, 6 juices and 6 finished products containing bilberry from different marketplaces and herbalists’ shops using LC-DAD-MS. A total of 15 anthocyanins were identified, and used for quality control purposes. Anthocyanidins were not detected in any of the samples analyzed. About 50% of these extracts differed significantly from the authentic bilberry, suggesting possible adulteration. About 60% of the extracts and 33% of the food supplements differed significantly from the authentic bilberry containing lower anthocyanin content than declared suggesting possible adulterations mainly with mulberry and chokeberry extracts. Gardana et al. [229] studied bilberry products from different producers using HPLC-DAD and FT-NIR/PCA for anthocyanins and their aglycones. Six of 71 products were adulterated with anthocyanins extracted from other berries (black mulberry, chokeberry, blackberry). The content of anthocyanins in these bilberry extracts was in the range from 18 to 34%. The same author in year 2020 [230] studied 17 cranberry commercial extracts and ten supplements. Based on cranberry markers and PCA, four extracts and six food supplements were not compliant; one extract seemed adulterated with *M. nigra* (black mulberry) and two with *Hibiscus* using UPLC-DAD-orbitrap-MS-PCA. Gasper et al. [231] investigated fresh/dried bilberries and dietary supplements using HPLC fingerprinting, LC-MS and TLC. The majority (91%) of dried bilberries from different sources were of good overall quality but dietary supplements revealed major problems, with 45% of unacceptable quality (e.g., bilberry-free products, nearly anthocyanin/tannin-free products, and samples being falsified with anthocyanins from other sources). Three bilberry juice samples were shown to have good quality.

Turbitt et al. analyzed cranberry products from different vendors using ^1^H-NMR and identified them to be adulterated with extracts from other plant species [41]. Mannino et al. studied cranberry extract products using HPLC-UV/Vis and orbitrap LC-MS, and found that 19 of 24 products were adulterated, possibly due to the misidentification of the raw materials [232]. Zhang et al. reported the adulteration of blueberry and cranberry juices with apple and grape juices at a 10% addition level using LC-QToF-MS based on eighteen characteristic markers and OPLS-DA predictive mode. The results demonstrated that metabolomics coupled with chemometric tools and global databases have potential as a reliable analytical method for food authentication [233].

Lee [234] studied *Vaccinium* supplements using HPLC and found that 14 of 45 products available as dietary supplements did not contain the fruit listed as ingredients. Six supplements contained no anthocyanins. Five others had contents differing from labeled fruit (e.g., bilberry capsules containing Andean blueberry fruit). Of the samples that did contain the specified fruit (n = 27), anthocyanin content ranged from 0.04 to 14.37 mg per 5.0 g of either capsule, tablet, or teaspoon. Lee analyzed Rosaceae (strawberry, cherry, blackberry, red raspberry, and black raspberry) dietary supplements (total n = 74) to determine their anthocyanin concentrations and profiles using HPLC-UV [235]. Lee studied black raspberry products and found seven of the 19 products contained no anthocyanins from black raspberry fruit using HPLC-DAD-MS. Five of the 33 dietary supplements contained no detectable anthocyanins, and another three were adulterated with anthocyanins from unlabeled sources [236].

Elderberries produce a small blue-black fruit typically made into jellies, jam, juice, wine or other processed products and rarely eaten fresh. Red and yellow fruits were generally not consumed; elderberry leaf, bark, and raw unripe fruits are suspected to contain cyanogenic glycosides as toxins and should not be consumed. The popularity of elderberry dietary supplements continued to rise during the COVID-19 pandemic due to their perceived immunomodulatory and antiviral activities. The surge in popularity of elderberry extracts and dietary supplements, combined with supply shortages and increased raw material costs, has created a situation in which some suppliers attempt to gain an unfair market advantage by selling low-quality or adulterated elderberry extracts. The most common adulterant appears to be black rice extract, but other unidentified materials are also used as adulterants. While the taxonomic status of American elderberry (*S. canadensis*) and European elderberry (*S. nigra*) is a matter of scientific debate, the two species can be distinguished based on their anthocyanin profiles. No mislabeling of European elderberry with American elderberry, or vice versa, was observed in commercial bulk extracts or finished products [237]. This report is based on data collected from elderberry suppliers, manufacturers, and contract analytical laboratories and provides evidence that adulteration of elderberry extracts is quite common and that suitable quality control methods are needed to properly authenticate elderberry extracts. Moreover, all processed elderberry samples as a juice, concentrate, natural colorant, and dietary supplements were produced from *S. nigra.* In another study using the UHPLC-PDA-MS method [182] involving 31 dietary supplements showed that more than 60% of the dietary supplements claiming to contain European black elderberry differed significantly from the authentic elderberry anthocyanin profile suggesting adulterations with black rice, purple carrot, and the flower of *S. nigra*. On the contrary, some papers suggest that elderberry extracts themselves have been used as adulterants, as undeclared color additives in wine or bilberry extracts [238,239].

Pomegranate fruit/juice is one of the most popular fruit juices, is well-known as a “superfood”, and plays an important role in healthy diets. Six anthocyanin pigments (major compounds (Dp-3,5-diglc, Dp-3-glc, Cy-3,5-diglc, Cy-3-glc), and minor ones (Pg-3,5-diglc, and Pg-3-glc)) are responsible for the red-purple color of pomegranate juice (PJ). Dasenaki et al. [240] developed a metabolomics approach using UHPLC-QToF-MS with chemometric tools for tracing the adulteration of pomegranate juice with apple and red grape juice. Zhang et al. [241] developed an international multi-dimensional authenticity specifications algorithm for distinguishing pure pomegranate juice and studied 45 juices from different manufacturers in the USA using HPLC-PDA. Krueger [242] studied the profiles of 793 pomegranate juices for authentication purposes and identified 477 juice samples that were found to be authentic based on consistent anthocyanin patterns and other compounds. Vardin et al. [243] applied FTIR spectroscopy in combination with a chemometric technique to detect the adulteration of pomegranate juice with grape juice. The main difference corresponds to C=O stretching found in the characteristic region of 1780–1685 cm^−1^ in IR spectroscopy. Pappalardo [244] studied the adulteration of pomegranate fruit juice with less expensive apple juice using UV-Visible spectroscopy with chemometrics. In another study, Türkyılmaz [245] showed a significant difference in the total monomeric anthocyanin contents using HPLC of pomegranate juice samples, ranging from 28 to 447 mg anthocyanins/L of juice. These differences may be due to variety or adulteration with grape, sour cherry, and apple juices.

Stoj A et al. [246], using an HPLC method, studied comparative analysis of juices from selected berries and found that the differences in anthocyanin composition will help detect adulterations of expensive raspberry and blackcurrant juices with less expensive strawberry and red currant juices. Pg-3-glc and Cy-3-xyl-rut were the main anthocyanins in strawberry and red currant juices, respectively, independent of variety. These were not identified in raspberry and blackcurrant juices, in which Cy-3-sop as well as Dp-3-rut and Cy-3-rut were the main anthocyanins.

Finally, developing innovative technical methods involving accuracy and lowering costs, should contribute to effectively resolving ongoing adulteration issues.

## 6. Health-Promoting Effects of Anthocyanins

Numerous in vitro, in vivo and clinical studies suggest that anthocyanins were positively implicated in human health. They exert different biological activities and suggested that the consumption of processed food or other formulations rich in these compounds may be correlated with antidiabetic [247,248], anti-obesity effects [247,249,250] as well as in preventing and inhibiting cancer growth [251,252,253]. Epidemiologic data substantiate the link between high consumption of anthocyanin-rich berries, fruits and vegetables and lowering the risk for cardiovascular disease [254]. Anthocyanins were reportedly involved in reduction of platelet aggregation in vitro, augmenting the endothelial function and conferring vascular protection. Anthocyanidins (aglycones of anthocyanins) extracted from *V. myrtillus* (bilberry) were reported to play a role in maintaining normal capillary structure [255]. Anthocyanins exhibited high antioxidant capacity which was reportedly involved in the induction of antioxidant enzymes, such as superoxide dismutase (SOD), catalase (CAT), and glutathione peroxidase (GPx) [256,257]. These enzymes are part of the endogenous defense system that protects cells from oxidative damage caused by ROS. Anthocyanins were also reported to activate the Nrf2–ARE signaling pathway, which reduces inflammation-associated disorders and atherosclerosis [258]. The neuroprotective effect of anthocyanins has been recently evaluated in different neural cell lines. Further, many anthocyanin-rich fruits comprising pomegranates, cranberries, cherries, and blueberries exhibited the ability to inhibit neuronal enzymes including monoaminoxidase A, acetylcholinesterase, and tyrosinase [1]. In connection to cognitive function, neuroprotective factors were recently identified and reduced the risk of cognitive impairment in aging [259]. Myriad evidence has shown the health benefits of anthocyanins on memory and cognition, as well as their neuroprotective effects in neurodegenerative disorders including Alzheimer’s disease [260].

## 7. Concluding Remarks

Anthocyanins occur prevalently in plants in combination with other polyphenols including different subclasses of flavonoids and phenolic acids. Additionally, the so-called anthocyanin-rich berries and fruits that were subjected to clinical trials for verifying their health-improving or medicinal effects most probably contain other polyphenolic compounds, though anthocyanins might be the major components. The coexisting polyphenols may interact in a synergistic or antagonistic way with anthocyanins and could serve as stabilizers or solubility enhancers. Thus, the outcomes of the clinical trials cannot be entirely attributed to the anthocyanins, and that adversely affects the rigor and integrity of these trials. Anthocyanin content and chemical composition in berries and fruit differ qualitatively and quantitatively from one species to another, and berries/fruit of the same species may show different content and chemical compositions if collected from different geographic regions.

The most widely used extraction solvents for anthocyanins are aqueous acetone, methanol, and ethanol. A variety of methods can achieve the qualitative and quantitative determination of anthocyanins. Currently, LC-DAD and LC-MS are the most popular and reliable methods for analyzing these compounds. The use of LC-DAD in combination with MS is an outstanding tool for the rapid separation, identification, and quantification of anthocyanins, taking advantage of the chromatographic and spectral characteristics of the LC system and the resolution and separation by mass fragments of MS. However, MS data do not give detailed and conclusive structural information, especially when isomeric compounds, such as *cis* and *trans* conformers, are studied. In such cases, the combination of both MS and NMR spectroscopies leads to unequivocal identification of the individual anthocyanins. NMR and MALDI-ToF-MS are very powerful and extremely sensitive techniques for the structure characterization of anthocyanins; however, both techniques are costly and require experienced and skilled operators, which limits their application and exploitation. In addition, integrating two or more methods, especially fingerprint comparison with HPLC and/or HPTLC analysis, can also improve the reliability of work and the quality aspect of berry/fruit products. Moreover, coupling spectroscopic and imaging tools with non-thermal methods is also an available option in future studies.

## Figures and Tables

**Figure 1 molecules-28-00560-f001:**
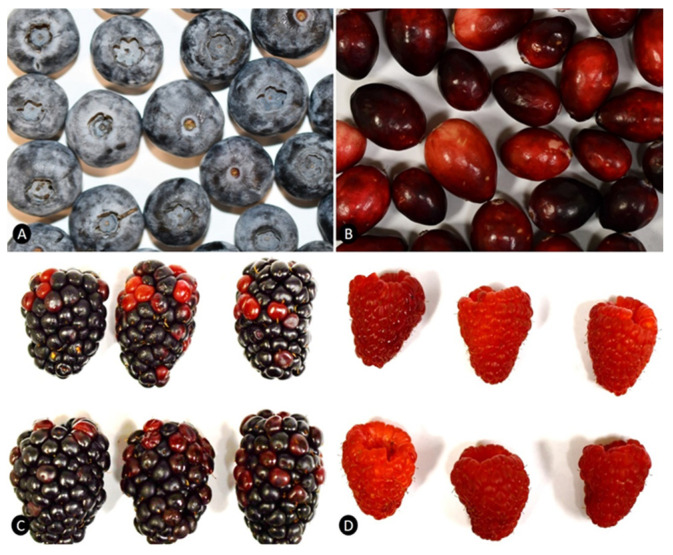
Types of berries. (**A**) berry (*Vaccinium myrtillus*). (**B**) False berry (*Vaccinium macrocarpon)*. (**C**,**D**) Aggregate drupelets (*Rubus fruticosus*) and *Rubus idaeus*.

**Figure 2 molecules-28-00560-f002:**
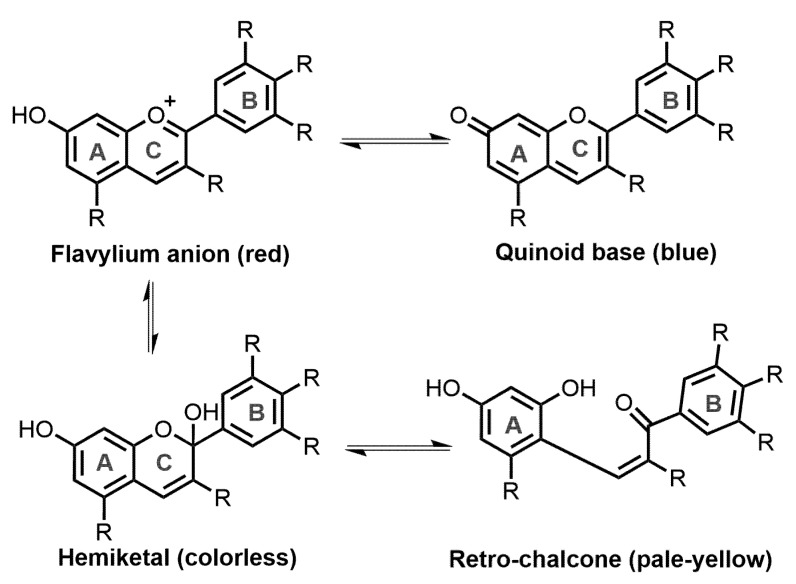
pH-Dependent structural isoforms of anthocyanins (A/B, aromatic rings and C, heterocyclic ring).

**Figure 3 molecules-28-00560-f003:**
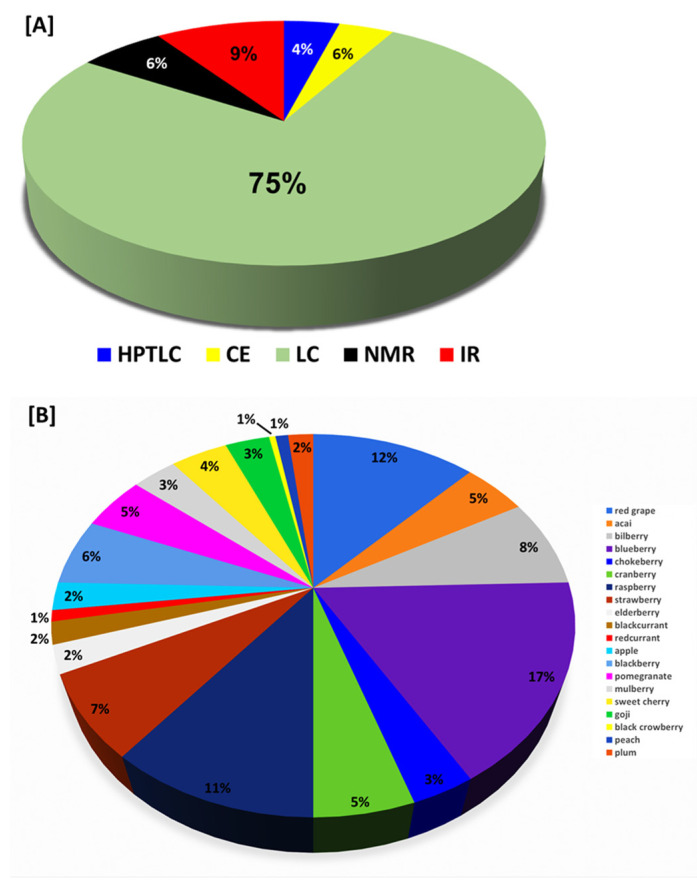
Published literature on anthocyanin analysis of fruits and berries between 2000 and 2022. (**A**) Contributions according to analytical methods. (**B**) Contributions according to the fruit/berries investigated.

**Table 1 molecules-28-00560-t001:** List of anthocyanin-containing berries and fruits.

No.	Botanical Name	Common Name	Family	Type
1	*Aronia melanocarpa* (Michx.) Elliott	Black Chokeberry	Rosaceae	Berry
2	*Euterpe oleracea* Mart.	Açaí Palm	Arecaceae	Drupe
3	*Fragaria × ananassa* (Duchesne ex Weston) Duchesne ex Rozier	Strawberry	Rosaceae	Aggregate drupelets
4	*Lycium ruthenicum* Murray	Black Goji Berry/Wolfberry	Solanaceae	Drupe
5	*Malus domestica* (Suckow) Borkh.	Apple	Rosaceae	Pome
6	*Morus nigra* L.	Black Mulberry	Moraceae	Aggregate drupelets
7	*Prunus avium* (L.) L.	Sweet Cherry	Rosaceae	Drupe
8	*Prunus domestica* L.	Plum	Rosaceae	Drupe
9	*Prunus persica* (L.) Batsch	Peach	Rosaceae	Drupe
10	*Punica granatum* L.	Pomegranate	Lythraceae	Berry
11	*Ribes nigrum* L.	Blackcurrant	Grossulariaceae	Berry
12	*Rubus fruticosus* Marshall	Blackberry	Rosaceae	Aggregate drupelets
13	*Rubus idaeus* L.	Raspberry (Red)	Rosaceae	Aggregate drupelets
14	*Rubus occidentalis* L.	Raspberry (Black)	Rosaceae	Aggregate drupelets
15	*Sambucus nigra* L.	Black Elderberry	Viburnaceae	Berry
16	*Vaccinium angustifolium* Aiton	Lowbush Blueberry	Ericaceae	Berry
17	*Vaccinium corymbosum* L.	Highbush Blueberry	Ericaceae	Berry
18	*Vaccinium macrocarpon* Aiton	Large Cranberry	Ericaceae	False berry
19	*Vaccinium oxycoccos* L.	Small Cranberry	Ericaceae	False berry
20	*Vaccinium myrtillus* L.	Bilberry	Ericaceae	False berry
21	*Vitis labrusca L.*	Concord Grapes	Vitaceae	Berry
22	*Vitis vinifera* L.	Grapes	Vitaceae	Berry

**Table 2 molecules-28-00560-t002:** Occurrence of anthocyanins in different berry and fruit varieties based on NMR and LC methods.

Compound	Açai	Goji Berry (Black) or *Black wolfberry*	Raspberry (Black and Red)	Strawberry	Cranberry (Small and Large)	Bilberry or Whortleberry	Blueberry (Highbush and Low Bush)	Black Elderberry	Mulberry (Black)	Blackberry	Chokeberry (Black)	Blackcurrant	Crowberry (Black)	Grapes (Red Peel)	Concord Grapes	Apples (Red Peel)	Sweet Cherry	Plum (Red and Black Peel)	Pomegranate Peel	Peach (Yellow Peel)
Cy-																				
3-*O*-gal					**+**	**+**	**+**		+		**+**		**+**			**+**		+		
3-*O*-glc	**+**	+	**+**	+	**+**	**+**	**+**	**+**	**+**	**+**	+	**+**	+	**+**	**+**	+	**+**	**+**	**+**	**+**
3-*O*-ara					**+**	**+**	+			+	**+**	+	**+**			+				
7-*O*-ara																+				
3-*O*-rut	**+**	+	**+**	+				+	**+**	+		**+**					**+**	**+**		+
3-*O*-sop			+																	
3-*O*-xyl			+							+	+					+		+		
3-*O*-sam	+		+					**+**												
3,5-*O*-diglc								+							+				**+**	
3-*O*-sam-5-*O*-rham			+																	
3-*O*-sam-5-*O*-glc								**+**												
3-*O*-malglc										+								+		
3-*O*-dioxlglc										+										
3-(6″-ace)glc															+			+		
3-(6″-*p*-cou)-5-diglc															+					
3-*O*-(6″-*p*-couglc)												+		+	+					
3-*O*-malglc-5-*O*-glc				+																
Dp-																				
3-*O*-gal						**+**	**+**					+	**+**							
3-*O*-glc		+			+	**+**	**+**					**+**	+	**+**	**+**				**+**	
3-*O*-ara					+	**+**	+						**+**							
3-*O*-rut												**+**								
3,5-*O*-diglc															+				+	
3-*O*-aceglc														+	+					
3-*O*-(6″-*p*-couglc)												+			+					
3-*O*-rut-(*p*-cou)-5-*O*-glc		+																		
Mv-																				
3-*O*-gal						**+**	**+**					+	**+**							
3-*O*-glc		+			+	**+**	**+**					+	+	**+**	**+**					
3-*O*-ara					+	**+**	+						**+**							
3-*O*-rut												+								
3,5-*O*-diglc															+					
3-*O*-aceglc														**+**	**+**					
3-*O*-*p*-couglc												+		**+**	**+**					
3-*O*-rut(*p*-cou)-5-*O*-glc		+																		
Pn-																				
3-*O*-gal					**+**	+	**+**					+	**+**							
3-*O*-glc	+	+			**+**	+	**+**					+	+	**+**	**+**		+	**+**		
3-*O*-ara					**+**	+	+						**+**							
3-*O*-rut	+		+	+								+					+	**+**		
3,5-*O*-diglc															+					
3-*O*-aceglc														+	+					
3-*O*-*p*-couglc														+	+					
Pt-																				
3-*O*-gal					+	**+**	**+**						**+**							
3-*O*-glc		+			+	**+**	**+**		+			+	+	**+**	**+**					
3-*O*-ara						**+**	+						**+**							
3-*O*-rut												+								
3,5-*O*-diglc															+					
3-*O*-rut-5-*O*-glc		+																		
3-*O*-aceglc														+	+					
3-*O*-*p*-couglc														+	+					
3-*O*-rut (*p*-cou)-5-*O*-glc		**+**																		
3-*O*-rut(fer)-5-*O*-glc		+																		
3-*O*-(*p*-cou)-rut		+																		
3-*O*-rut(caf)-5-*O*-glc		+																		
Pg-																				
3-*O*-gal				+	+						+									
3-*O*-glc	+	+	+	**+**				+	**+**			+		+	+			+	+	
3-*O*-ara				+	+						+									
3-*O*-rut			+	+					+			+					+			
3-*O*-sam								+												
3-*O*-soph			+																	
3,5-*O*-diglc				+															+	
3-*O*-aceglc				+																
3-*O*-malglc				+																

Note: (Pg) pelargonidin; (Cy) cyanidin; (Dp) delphinidin; (Pt) petunidin; (Pn) Peonidin; (Mv) malvidin; (glc) glucoside; (rut) rutinoside; (gal) galactoside; (rham) rhamnoside; (soph) sophoroside; (diglc) diglucoside; (xyl) xyloside; (gal) galactoside; (ara) arabinoside; (sam) sambubioside; (mal) malonyl; (*p*-cou) *p*-coumaroyl; (caf) caffeoyl; (fer) feruloyl; (ace) acetyl. Anthocyanin composition is cultivar-dependent, for, e.g., two blue berry cultivars (‘K78-16’ and ‘416’) also contain 11 acetylated anthocyanins (Dp-3-*O*-acelglc, Dp-3-*O*-acegal, Cy 3-*O*-aceglc, Cy-3-*O*-acegal, Cy-3-*O*-aceara, Pt-3-*O*-aceglc, Pt-3-*O*-acegal, Pn-3-*O*-aceglc, Pn-3-*O*-acegal, Mv-3-*O*-aceglc, Mv-3-*O*-acegal) and others no acylated anthocyanins. Anthocyanin profiles of blueberries differ from those of bilberries with respect to the high proportions of Mv-3-ara (24%), Dp-3-gal (20%), and Pt-3-gal (16%) and low levels of Cy-3-glc, Pt-3-glc, Pn-3-glc, Pn-3-ara, and Mv-3-glc. Although the anthocyanin content in berries or fruits can change and degrade with processing, harvesting time and storage, anthocyanin profile will be unique, and its qualitative pattern is characteristic. Acylated berries anthocyanins are described in the literature as minor components only. However, they may be of great help to identify the berry source of anthocyanins. Across the blueberry species and genotypes, malvidin and delphinidin aglycones accounted for majority of anthocyanin glycosides. Red highlighted ones are either markers or major compounds.

**Table 5 molecules-28-00560-t005:** Total anthocyanin content of commonly consumed berries and fruits using analytical methodologies.

Source	Content * (mg/100 g)	References
Açai (*Euterpe oleracea* Mart.)	48.9–303.7 (FW) 10–3410 (DEW)	[84,88,91,93,95,96]
Black Goji berry or black wolfberry (*Lycium ruthenicum* Murray)	930–5826 (DW) 343–525 (FW) tr–380 (DEW)	[148,149,183,184,198]
Raspberry (red) (*Rubus idaeus*)	0.1–133.9 (FW) 76.2–365.2 (DW) 336–1030 (DEW)	[89,97,102,105,113,132,152,153,159,160,161]
Raspberry (black) (*Rubus occidentalis*)	381–541.3 mg/100 mL puree 2885–11109 (DEW)	[132,150]
Strawberry (*Fragaria* × *ananassa*)	0.4–84 (FW) 97.5–758.6 (DW) 520 (DEW)	[97,105,113,136,151,152,153,166,167,168,171,172,173]
Cranberry (Small and large)		
American large cranberry (*Vaccinium macrocarpon* Aiton)	78–430 (DW)	[189]
European small cranberry (*Vaccinium oxycoccus* L)	40.7–360 (FW) 397 (DW)	[105,106,162]
Bilberry (*Vaccinium myrtillus*)	104–2025.6 (FW) 2298–7465 (DW)	[87,92,100,105,107,113,120,204]
Blueberry (highbush and low bush)	77.5–820 (FW) 558.3–1188.3 (DW) 1020 (DEW)	[83,97,102,113,136,153]
lowbush blueberry (*Vaccinium angustifolium* Aiton)	110–725 (FW) 2400 (DEW)	[106,109,110]
highbush blueberry (*Vaccinium corymbosum* L.)	42–6720 (FW) 30–2800 (DEW)	[108,112,115,116,118,120,129,138]
Black Elderberry (*Sambucus nigra* L.)	391–1816 (FW)	[147,163,182,216]
Black Mulberry (*Morus nigra*)	125.3–1610 (DEW)	[113,146]
Blackberry (*Rubus fruticosus* L.)	97.7–647 (FW) 192.4–1010 (DW) 547–27230 (DEW)	[83,86,89,97,99,101,102,113,132,202]
Black Chokeberry (*Aronia melanocarpa* Elliott)	249–737 (FW) 1041 (DW)	[105,155]
Blackcurrant (*Ribes nigrum*)	130–586.6 (FW) 756–1064 (DW) 2100 (DEW)	[105,107,113,118]
Black crowberry (*Empetrum nigrum*)	401–768.2 (FW) 2473–4180 (DW)	[105,113,156]
Grapes (red peel) (*Vitis vinifera* L.)	2.4–710 (FW) 700–56470 (DEW)	[79,157,165,185,186,187]
Apples (red peel) (*Malus domestica*)	11–26.1 (FW)	[85,90]
Sweet cherry (*Prunus avium* L.)	0.3–642.5 (FW)	[140,141,142,144,145,202]
Plum (red and black peel) (*Prunus domestica*)	12.9–161.4 (FW)	[199]
Pomegranate (*Punica granatum* L. Cv Ermioni)	nd–344.1 (FW) 7.9–44.7 (DW)	[175,179,180,201,207,208]
Peach (yellow peel) (*Prunus persica*)	0.5–33.6 (FW)	[199,205]

* FW: fresh weight; DW: dry weight; DEW: dry extract weight. The variations in yields might be due to geographic location, environmental factors, genotypes, collection season, cultivar, ripening stage, processing, and storage conditions. High concentrations of anthocyanins were present in the peel, in some also in the fruit flesh (e.g., bilberry), and ripened fruit as the accumulation of anthocyanins begins at the onset of ripening. Most of these total anthocyanins were expressed as Cy-3-glc equivalents.

**Table 6 molecules-28-00560-t006:** Summary of analytical methods for determining the adulteration of fruit/berries and fruit/berries food-based products concerning anthocyanins.

No.	Fruit/Berry	Type	Analytical Method	Chemometrics	Adulterants/Contaminants	Ref
1	Bilberry	Extract	HPLC-MS, NMR	−	Amaranth/Red dye#2	[225]
2	Bilberry	Supplements	HPLC-UV-MS	−	-	[226]
3	Bilberry	Supplements	HPLC-UV-MS	−	-	[227]
4	Bilberry	Supplements	HPLC-DAD, FT-NIR/PCA	+	Black mulberry, chokeberry, blackberry	[229]
5	Cranberry	Extracts and supplements	UPLC-DAD-MS	+	Black mulberry, Hibiscus	[230]
6	Bilberry	Fresh/Dried and Dietary supplements	LC-MS and TLC	−	-	[231]
7	Cranberry	Supplements	^1^H-NMR	−	-	[41]
8	Blueberry/ cranberry	Juices	LC-QToF-MS	+	Apple and grape juices	[233]
9	Bilberry	Supplements	HPLC-UV	−	Andean blueberry fruit	[234]
10	Strawberry, Cherry, Blackberry, Red raspberry	Supplements	HPLC-UV	−	-	[235]
11	Black raspberry	Supplements	HPLC-DAD-MS	−	Blackberry	[236]
12	Black elderberry	Supplements	UHPLC-PDA-MS	−	Black rice, purple carrot and flower of *S. nigra*	[182,237]
13	Red grape	Wine	HPLC	−	Elderberry	[238]
14	Pomegranate	Juice	UPLC-QToF	+	Apple and red grape juice	[240]
15	Pomegranate	Juice	HPLC-PDA	−	Apple, pear, grape, sour cherry, plum, or Aronia juice	[241]
16	Pomegranate	Juice	HPLC-PDA	−	Apple, Pear, Cherry, Grape, Blackberry, Grape skin or Aronia	[242]
17	Pomegranate	Juice	FTIR	+	Grape juice	[243]
18	Pomegranate	Juice	UV-Vis	+	Apple juice	[244]
19	Pomegranate	Juice	HPLC	−	Grape, Sour cherry and apple juice	[245]
20	Raspberry and Blackcurrant	Juice	HPLC-DAD	−	Strawberry and Redcurrant juices	[246]

## Data Availability

The data presented in this study are available on request from the corresponding author.

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
