# Peer review of "Advances in the Chemistry, Analysis and Adulteration of Anthocyanin Rich-Berries and Fruits: 2000–2022"

_molecules, 2023, doi:10.3390/molecules28020560_

Round 1

Reviewer 1 Report

This is an exhaustive (both in content and, to some extent, for the reader) but useful review of more recent work on analysis of anthocyanins in fruits. The English is quite good and the review touches on a series of related topics including adulteration of anthocyanin-containing products and health benefits of anthocyanin consumption.

Minor points are:

p. 5 line 162 and Figure 2: flavylium cation, not anion.

p. 10 line 372 and many other places in the text and tables:  the name of the fruit is not acai, but rather açai, a three-syllable word with a cedilla instead of the letter “c” that is pronounced ah-sah-EE.

Author Response

Minor points are:

  1. 5 line 162 and Figure 2: flavylium cation, not anion: Corrected
  2. 10 line 372 and many other places in the text and tables: the name of the fruit is not acai, but rather açai, a three-syllable word with a cedilla instead of the letter “c” that is pronounced ah-sah-EE. Corrected

Reviewer 2 Report

This review has mainly focuses on various analytical methods that are used for characterization, quantification and chemical analysis of anthocyanins present in fruits and vegetables in last two decades. Authors have mentioned that this review is more of a compiling various analytical methodologies, I think it is important to understand the suitable technique for future researchers who work on anthocyananis. Given the volumes of publications raise in last two decades on fruits and vegetables, especially related to their health benefits, major claims have been attributed towards anthocyanins and it is essential to understand the quality and quantity of anthocyanins present in them using appropriate analytical technique. 

Overall this review is well drafted. However, there is an improper and extensive use of definite article "the" in text. 

Increase the quality of Table2. 

Author Response

Overall this review is well drafted. However, there is an improper and extensive use of definite article "the" in text.  Checked and corrected

Increase the quality of Table 2. New table added

Reviewer 3 Report

It is very interesting for have a full sight on the the cehmistry et al of antho-cyanin in fruits for last 20 years. the review organized wery well and is very useful to people who want to know the advance knoledge of pigment in berry or fruits. one suggestion is that the table 4 is so long that should be dealt as an attachement of the review paper.

Author Response

Reviewers Comment-3:

It is very interesting for have a full sight on the the cehmistry et al of antho-cyanin in fruits for last 20 years. the review organized wery well and is very useful to people who want to know the advance knoledge of pigment in berry or fruits. one suggestion is that the table 4 is so long that should be dealt as an attachement of the review paper. Table 4 could not be attached as there is no way to do it.